# An intranasal subunit vaccine induces protective systemic and mucosal antibody immunity against respiratory viruses in mouse models

Aina Karen Anthi [1,2,3], Anette Kolderup[1,2,3,12], Eline Benno Vaage[1,3,4,12], Malin Bern[1,2,12], Sopisa Benjakul [1,2,3], Elias Tjärnhage [1,4], Fulgencio Ruso-Julve [1,2,3], Kjell-Rune Jensen[1,2,3], Heidrun Elisabeth Lode [1,2,3,5], Marina Vaysburd[6], Jeannette Nilsen[1,2,3], Marie Leangen Herigstad[1,2,3], Siri Aastedatter Sakya[1,2,3], Lisa Tietze [1,3], Diego Pilati [7], Mari Nyquist-Andersen [1,2,3], Mirjam Dürkoop[1,2,3], Torleif Tollefsrud Gjølberg [1,2,3,5], Linghang Peng[8], Stian Foss[1,2,3], Morten C. Moe [5], Benjamin E. Low [9], Michael V. Wiles[9], David Nemazee [8], Frode L. Jahnsen[4,10], John Torgils Vaage[1], Kenneth A. Howard [7], Inger Sandlie[11], Leo C. James [6], Gunnveig Grødeland [1,4], Fridtjof Lund-Johansen [1,3] & Jan Terje Andersen [1,2,3] ✉

Although vaccines are usually given intramuscularly, the intranasal delivery route may lead to better mucosal protection and limit the spread of respiratory virus while easing administration and improving vaccine acceptance. The challenge, however, is to achieve delivery across the selective epithelial cell barrier. Here we report on a subunit vaccine platform, in which the antigen is genetically fused to albumin to facilitate FcRn-mediated transport across the mucosal barrier in the presence of adjuvant. Intranasal delivery in conventional and transgenic mouse models induces both systemic and mucosal antigen-specific antibody responses that protect against challenge with SARS-CoV-2 or influenza A. When benchmarked against an intramuscularly administered mRNA vaccine or an intranasally administered antigen fused to an alternative carrier of similar size, only the albumin-based intranasal vaccine yields robust mucosal IgA antibody responses. Our results thus suggest that this needle-free, albumin-based vaccine platform may be suited for vaccination against respiratory pathogens.

Respiratory infectious diseases, such as COVID-19 and influenza, have a disrupting impact on society. While intramuscularly administered vaccines are effective against severe disease and death, limited immune responses are induced locally at the site of infection, which affect their ability to reduce viral transmission[1–7]. The mucosal immune system represents the first line of defense against invading pathogens at epithelial cell surfaces. As the nasal cavity and the lungs are the primary entry points for many viruses, delivery of vaccines via the mucosal surfaces offers an ideal approach for obtaining both protective immunoglobulin G (IgG) and secretory immunoglobulin A (SIgA)

responses at these sites[4,8–11]. Antibodies have been detected at mucosal sites of both animals and humans after intramuscular vaccination, but mainly of the IgG isotype, while exposure to SARS-CoV-2 also raises robust IgA responses in the respiratory tract[12–15]. SIgA antibodies, present in the upper and lower respiratory tract, are known to provide immediate immunity by eliminating pathogens at the mucosal barrier. This is possible as SIgA engages the polymeric immunoglobulin receptor (pIgR) on the basolateral side of mucosal epithelial cells for directional transport across the barrier[4,11]. For instance, it has been demonstrated that SIgA is pivotal in providing immunity in the upper respiratory tract following influenza virus infections[16]. Regarding COVID-19 patients, elevated levels of IgA antibodies are detected in the saliva and nasal secretions following the onset of symptoms[8,17,18]. In contrast, intramuscular mRNA vaccination alone induces minimal mucosal SIgA responses in individuals without pre-exposure to SARS-CoV-2[18], while such responses are more efficient in combination with natural infection[8,9,14,18].

As such, there is an increasing interest in development of vaccine technologies that allow the induction of protective antibody responses not only systemically but also at the actual site of infection. A range of pre-clinical and clinical efforts are ongoing[5,6], where, for instance recent studies in non-human primates show promising results for adenovirus-based SARS-CoV-2 vaccines[6,19–21]. So far, only a handful has obtained approval or Emergency Use Listing domestically[5,6,22], including the SARS-COV-2 spike protein encoding adenovirus-vectored formulations from CanSinoBIO (Convidecia Air) and Baharat Biotech (iNCOVACC)[5,22,23]. Another example is a live attenuated influenza vaccine (Flumist/Fluenz Tetra) that is approved for intranasal delivery to healthy individuals under the age of 50 in the US and Europe, though the European Medicines Agency has restricted vaccination to healthy children aged 2-18 years due to limited effect in adults[23]. None of these strategies aim for targeting of cellular receptors as a gateway for transmucosal transport, and delivery to mucosa-associated lymphoid tissue has not yet been fully explored.

To meet this challenge, we have developed a subunit vaccine platform that targets the neonatal Fc receptor (FcRn), which is expressed by polarized epithelial cells lining the respiratory mucosal surfaces[24–27]. Here, FcRn mediates transport of IgG antibodies and immune complexes bidirectionally across the barriers, which occurs in a strictly pH-dependent manner, binding in mildly acidic endosomes followed by release at the cell surface at neutral pH[25,28,29]. Furthermore, FcRn also engages albumin via a binding site that is distal and non-overlapping with that of IgG, facilitating simultaneous binding of the two ligands[30–33]. Studies have shown FcRn-directed transport of IgG and IgG Fc-fused antigens[34–37], as well as transport of albumin and albumin-fused antigens across epithelial cell barriers may lead to induction of antigen-specific antibody responses[38–40]. This is highly interesting, as we have demonstrated that human serum albumin (HSA) is transported more efficiently than human IgG across mucosal surfaces after intranasal administration to human FcRn-expressing mice[33]. Importantly, delivery can be further enhanced using HSA engineered for improved FcRn binding properties, mediating increased binding at acidified pH without affecting binding at neutral pH[33].

Here, we describe the use of an engineered albumin with improved pH-dependent FcRn-binding properties as a carrier for respiratory viral antigens. Intranasal delivery of the designed albumin-fused subunit vaccines, in the presence of adjuvant, demonstrates robust induction of systemic antigen-specific IgG responses, but importantly, also both antigen-specific IgG and IgA responses locally at the mucosal surfaces. Moreover, the vaccinated mice show complete protection against a lethal challenge with the respective viruses. Thus, engineered albumin-fused subunit vaccines targeting mucosal FcRn should be an attractive needle-free vaccination strategy to achieve both systemic and mucosal protective antibody immunity.

## Results and discussion

To explore this approach, we first genetically fused the receptor-binding domain (RBD) from the ancestral SARS-CoV-2 (Wuhan) to mouse serum albumin (MSA) (RBD-MSA) (Fig. 1a). This fusion, produced in large amounts as a pure monomer (Supplementary Fig. 1a), bound strongly to the SARS-CoV-2 host receptor, angiotensin-converting enzyme 2 (ACE2) (Supplementary Fig. 1b) as well as mouse FcRn (Supplementary Fig. 1c). Importantly, the structural integrity of RBD fused to MSA was further confirmed as it showed similar reactivity to three clinically approved anti-SARS-CoV-2 antibodies (sotrovimab, cilgavimab and tixagevimab) as RBD alone (Fig. 1b). Next, intranasal vaccination was performed in BALB/c mice using CpG as an adjuvant, and RBD-MSA was compared with equimolar amounts of unfused RBD. Three weeks later a booster dose (10% of prime) was given. Blood samples were collected throughout the study, and bronchoalveolar lavage fluid (BALF) was collected at the endpoint, at week 5 (Fig. 1a). Using a high-throughput flow cytometry bead-based method detecting RBD-specific antibodies by fluorescently labeled secondary antibodies[41,42], we demonstrated induction of robust RBD-specific IgG responses after intranasal vaccination with RBD-MSA, in both serum and BALF samples, compared to close to undetectable levels for unfused RBD (Fig. 1c and Supplementary Fig. 2a, b). Furthermore, robust levels of RBD-specific IgA were detected in BALF, and only in BALF from mice vaccinated with RBD-MSA (Fig. 1c and Supplementary Fig. 2b). Thus, mucosal FcRn-targeting was needed to mount the RBD-specific IgG and IgA antibody responses.

To assess the ability of the generated antibodies to inhibit binding of antigen to the SARS-CoV-2 host receptor ACE2, RBD-coupled beads were incubated with serum or BALF samples in the presence of recombinant human ACE2 conjugated to digoxigenin (DIG), that was detected with a phycoerythrin (PE)-conjugated DIG-specific secondary antibody[42] (Fig. 1d). Only sera and BALF from mice vaccinated with RBD-MSA contained RBD-specific antibodies efficiently inhibiting ACE2 binding (Fig. 1d). In a neutralization assay, 293T cells expressing both human ACE2 and the transmembrane protease serine 2 (TMPRSS2) (293T-hACE2-TMPRSS2) were exposed to a replication deficient SARS-CoV-2 pseudotyped lentivirus expressing GFP, allowing for quantification of cellular infection by GFP expression[43], and thus the ability of vaccine-induced antibodies to block infection (Fig. 1e). The results revealed efficient blocking of cellular infection, but only in the presence of sera and BALF from mice intranasally vaccinated with RBD-MSA (Fig. 1e).

To evaluate the protective capacity of the induced antibody responses, we took advantage of mice transgenic for human ACE2 (K18-hACE2) (Fig. 2a) to perform a SARS-CoV-2 challenge study after intranasal vaccination with RBD-MSA or RBD alone as above. Again, we observed that the albumin fusion design induced strong systemic anti-RBD antibody responses (Fig. 2a and Supplementary Fig. 2c), which blocked ACE2-RBD binding and cellular infection (Fig. 2b). The mice were then challenged with live SARS-CoV-2 WA1/2020 at week five, demonstrating no loss of weight for mice vaccinated with RBD-MSA, while those vaccinated with RBD alone or unvaccinated (PBS) showed a significant weight reduction (Fig. 2c). Furthermore, mice vaccinated with RBD-MSA had significantly lower levels of viral load in the lungs at day seven post infection (Fig. 2d), and lower level of inflammation which reflects limited lymphocyte infiltration of the hematoxylin and eosin-stained lung tissue (Fig. 2e).

Next, we addressed how our intranasal albumin-based vaccine strategy compares with intramuscular vaccination with an mRNA strategy, with a particular focus on the induction of mucosal antibody immunity. This was done by vaccinating mice intramuscularly with the COVID-19 BioNTech-Pfizer mRNA vaccine (Comirnaty/BNT162b2) in parallel with RBD-MSA followed by a booster dose (10% of prime) for both strategies, and the harvest of blood and mucosal flushes from the

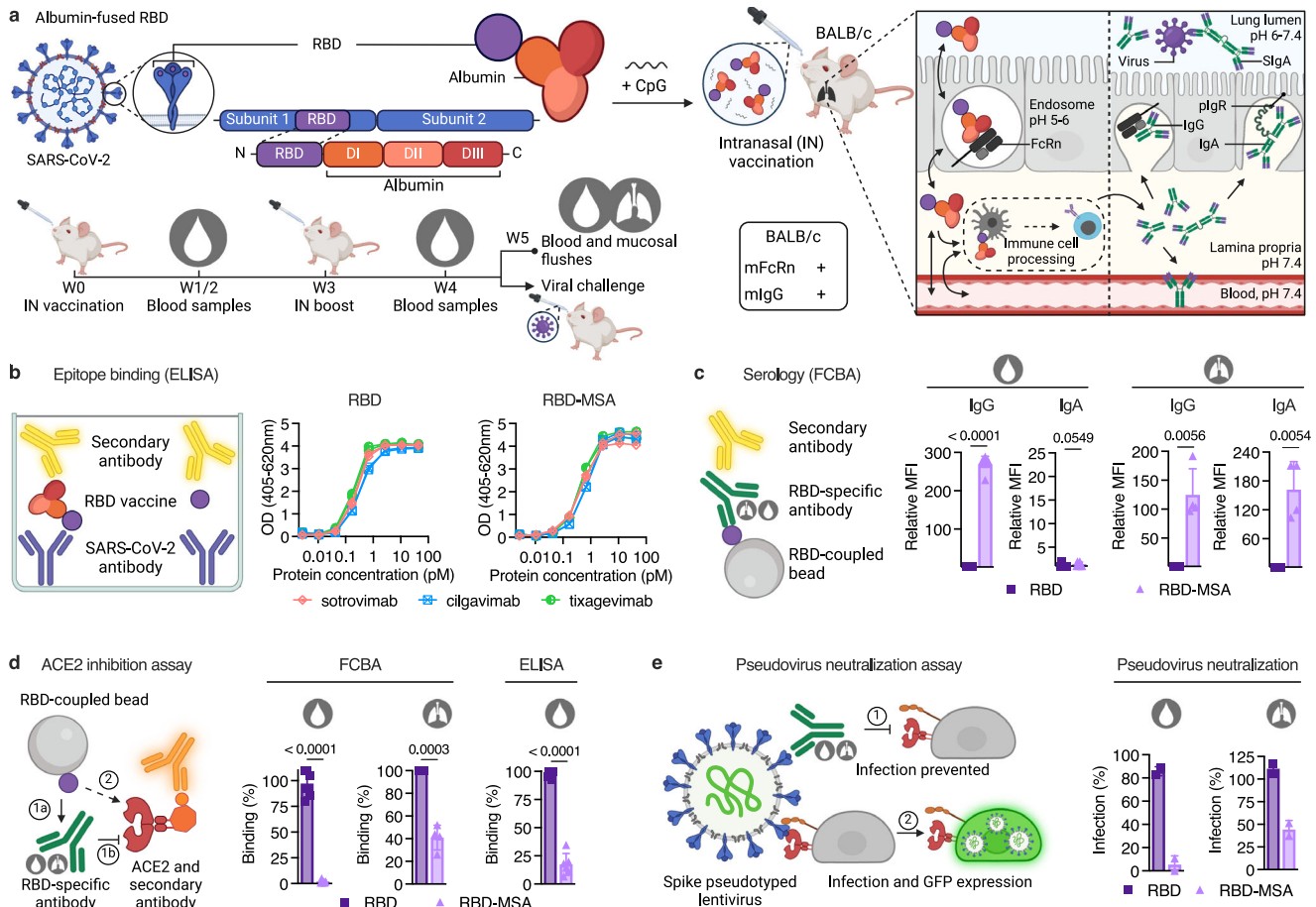

**Fig. 1 | Intranasal vaccination with RBD-fused MSA in female BALB/c mice induces protective antigen-specific IgG and IgA antibodies. a** Top panel: Illustration of a fusion design with albumin genetically fused to RBD, and intranasal vaccination followed by FcRn-mediated transport of the vaccine across polarized mucosal epithelial cells, immune cell processing, and transport of generated IgG and IgA antibodies by FcRn and the pIgR, respectively, to the mucosal surface, and also blood for IgG. Bottom panel: Flow chart outlining the intranasal vaccination protocol with live virus challenge or endpoint sample harvest at week 5. Mice were vaccinated with equimolar amounts of RBD and RBD-MSA (prime dose: 6.2 μg and 19.9 μg, respectively, boost: 10% of prime) in combination with 20 μg CpG, or given PBS. **b** Illustration of ELISA and data for determination of RBD epitope availability on the RBD-albumin fusion compared to RBD alone, using the commercial monoclonal antibodies sotrovimab, cilgavimab and tixagevimab. **c** Illustration of the flow cytometric bead array (FCBA) for detection of antigen-specific antibodies in sera and BALF and data from analyses at endpoint. **d** Illustration of the FCBA for detection of antibodies able to inhibit human ACE2-RBD binding. Antibodies can inhibit human ACE2 binding to RBD-coupled beads (1a-1b), or not, leading to detection of ACE2-DIG bound to RBD-coupled beads by anti-DIG-PE (2), and results from sera and BALF at endpoint, detected using FCBA and ELISA (only sera). **e** Illustration of a pseudovirus neutralization cellular assay based on lentiviral infection of 293T-hACE2-TMPRSS2 cells where the presence of antibodies blocks cellular infection (1), while reduced blocking ability leads to pseudoviral entry and GFP expression (2), and results from sera and BALF at endpoint using SARS-CoV-2 pseudovirus of the ancestral strain (Wuhan). Curve plots in **b** are presented as each replicate of technical duplicates. Bars indicating **c**, **d** group mean ± SD with individual mice represented as a single datapoint (sera: *n* = 6, BALF: RBD *n* = 3 and RBD-MSA *n* = 4) and **e** technical duplicates of pooled biological samples per group (sera: *n* = 6, BALF: RBD *n* = 3 and RBD-MSA *n* = 5). **c**, **d** Two-tailed unpaired t-test. **a** Created or **b**–**e** partially created in BioRender[86].

upper and lower respiratory and distal mucosal sites (Fig. 3a). The results revealed induction of robust RBD-specific IgG antibody responses for both strategies, systemically as well as in BALF, where intranasal vaccination gave at least as good responses as intramuscular mRNA (Fig. 3b and Supplementary Fig. 3a). Interestingly, the mRNA strategy induced higher titers of IgG in saliva and nose secretion (Fig. 3b). Furthermore, serum and BALF samples from both RBD-MSA and BioNTech-Pfizer vaccinated mice blocked the interaction between ACE2 and RBD (Fig. 3c). In contrast, only intranasal vaccination with RBD-MSA gave rise to RBD-specific IgA responses in the respiratory tract and at distal mucosal sites (Fig. 3b and Supplementary Fig. 3b, c). Also, flow cytometry analyses of cells from the harvested mediastinal lymph nodes revealed the highest percentage of RBD-specific B-cells in mice vaccinated with RBD-MSA (Fig. 3d and Supplementary Fig. 4). No difference was seen in the frequency of RBD-specific B-cells between mice receiving BioNTech-Pfizer or PBS, which is as expected as the

mediastinal lymph node is not the draining lymph node after intramuscular injection with an mRNA vaccine[44] (Fig. 3d).

The results motivated further exploration of albumin as a vaccine subunit carrier in a human context. Importantly, we have demonstrated that there are significant differences in ligand:FcRn binding affinity and specificity between mice and man[45–47], knowledge which must be taken into consideration when designing albumin-based strategies. Thus, we took advantage of an engineered HSA variant (QMP) harboring three amino acid substitutions (E505Q/T527M/K573P) that improve binding to human FcRn at acidic pH without affecting release at neutral pH[33]. The rationale for this choice is that we have demonstrated that QMP can extend the plasma half-life of fusion products by up to 3.5-fold in human FcRn transgenic mice[33,48], and also enhance delivery across mucosal barriers of such mice after needle-free intranasal delivery[33,49]. The RBD-QMP fusion was produced as a pure monomeric protein with the expected

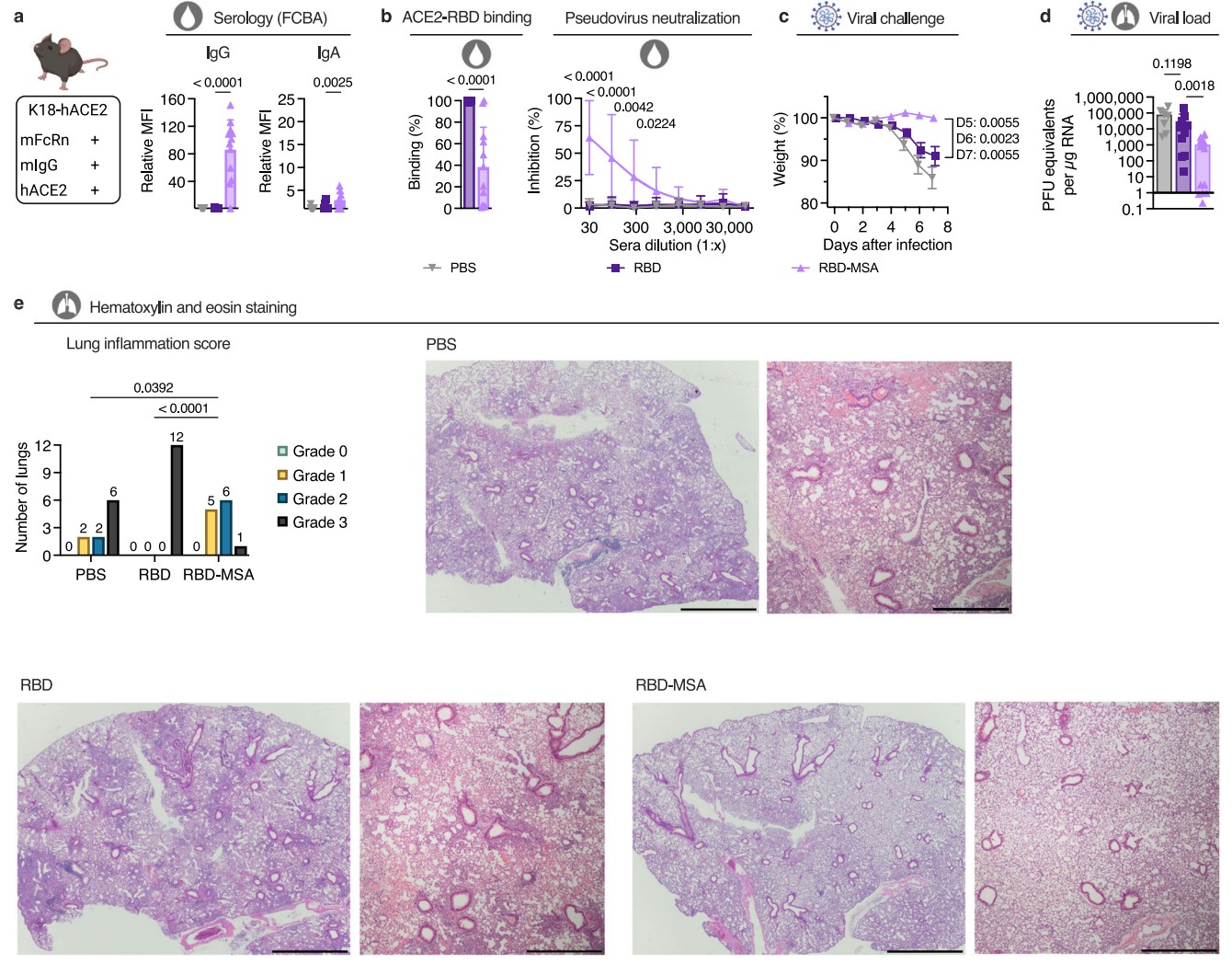

**Fig. 2 | Intranasal vaccination with RBD-fused MSA protects female K18-hACE2 mice from lethal challenge with SARS-CoV-2. a** Illustration of the K18-hACE2 mouse model and RBD-specific antibodies in sera at day 28 detected using FCBA. **b** The ability of antibodies in sera of vaccinated K18-hACE2 mice at day 28 to inhibit ACE2-RBD binding and block cellular infection by SARS-CoV-2 pseudovirus of the ancestral strain (Wuhan). **c** Weight change after viral challenge of intranasally vaccinated K18-hACE2 mice by SARS-COV-2 WA1/2020. **d** Viral load in the lungs of intranasally vaccinated K18-ACE2 mice 7 days post infection **e** Hematoxylin and eosin staining of lungs of intranasally vaccinated K18-ACE2 mice 7 days post infection. One representative lung from each group is shown with magnification x2 (scale bar 1 mm) and x4 (scale bar 500 μm). **a, b, d** Bar indicating group mean ± SD

with individual mice represented as a single datapoint (PBS *n* = 10, RBD and RBD-MSA *n* = 12, except **d** PBS *n* = 7)). **b** Curve plot presented as group mean ± SD of technical duplicates (PBS *n* = 10, RBD and RBD-MSA *n* = 12). **e** Bars indicating number of biological samples per indication within each treatment group (group sizes: PBS *n* = 10, RBD and RBD-MSA *n* = 12). Weight in **c** presented as percentage weight compared to the weight at day of infection, as biological group mean ± SEM (PBS *n* = 10, RBD and RBD-MSA *n* = 12). **a** and **c** One-way ANOVA with Tukey's multiple comparison, **b** Two-tailed unpaired t-test (ACE2-RBD binding) or Kruskal-Wallis test with Dunn's multiple comparison (pseudovirus neutralization), and **d-e** Two-tailed Mann-Whitney test. **a**–**e** Partially created in BioRender[86].

molecular weight (Supplementary Fig. 5a) and bound human ACE2 (Supplementary Fig. 5d), in line with similar binding responses to epitope specific SARS-CoV-2 antibodies as RBD alone (Fig. 1b and Supplementary Fig. 5e). In addition, RBD-QMP bound human FcRn more strongly at acidic pH than RBD fused to wild-type HSA (RBD-WT) (Supplementary Fig. 5f). RBD-QMP was more efficiently recycled in a human endothelial cell-based recycling assay (HERA) (Supplementary Fig. 5g), and also transcytosed across polarized epithelial cells expressing human FcRn better than RBD-WT (Supplementary Fig. 5h).

Pre-clinical in vivo studies of HSA-based designs have been performed in mice transgenic for human FcRn, as we have demonstrated that mouse FcRn engages HSA very poorly[45–47], and thus, the mouse receptor will ignore HSA in the presence of high amounts of endogenous MSA. To make it even more complex in a translational

perspective, mouse IgG subclasses have a short plasma half-life in human FcRn transgenic mice, as they only interact weakly with the receptor[45,50,51]. We therefore used a mouse strain made transgenic for chimeric IgG1 with a human Fc-region (chIgG1) as well as human FcRn (Tg32-hFc)[52] (Fig. 4a). The Tg32-hFc mice were vaccinated intranasally with equimolar amounts of RBD-QMP or unfused RBD, or given PBS, following the same vaccination regimen as for the BALB/c mice (Fig. 3a). RBD-QMP induced robust titers of RBD-specific IgG both systemically and mucosally, as well as IgA at local and distal sites (Fig. 4a and Supplementary Fig. 6a–c). We also observed that the antibodies in serum and BALF samples blocked ACE2-RBD binding (Fig. 4b) and prevented cellular infection (Fig. 4c). In addition, the percentage of the total B-cell population in the mediastinal lymph nodes that was RBD-specific was greatly increased in the mice intranasally vaccinated with RBD-QMP (Fig. 4d).

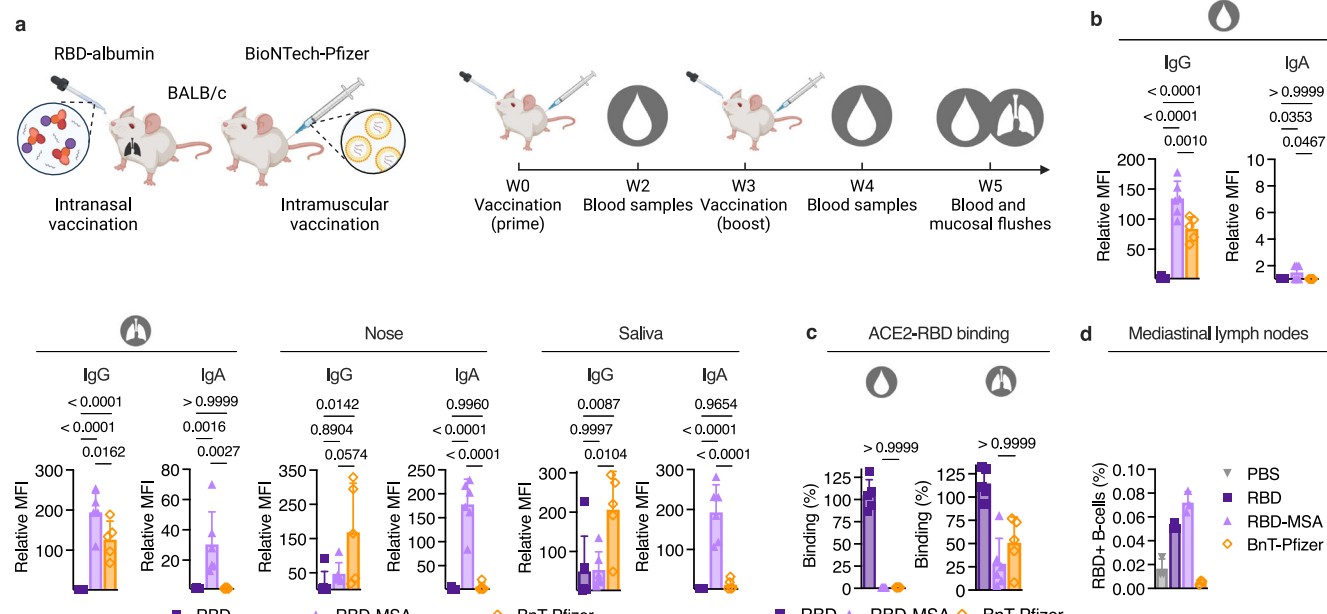

**Fig. 3 | Intranasal vaccination with RBD-fused MSA in female BALB/c mice induces robust antigen-specific antibodies beyond that of intramuscular vaccination. a** Illustration of intranasal vaccination with an RBD-albumin fusion compared with intramuscular vaccination with the mRNA vaccine from BioNTech-Pfizer (Comirnaty/BNT162b2), and the vaccination regimen in female BALB/c mice. **b** RBD-specific IgG and IgA in sera and BALF at endpoint after vaccination of BALB/c mice (prime dose: 6.2 μg RBD and 19.9 μg RBD-MSA in combination with 20 μg CpG, 3 μg Comirnaty, or PBS), and **c** the ability of the antibodies to inhibit human ACE2-RBD binding. **d** Percentage of B-cells that are antigen-specific in mediastinal lymph nodes detected by flow cytometry. All serology detected using FCBA. Bars indicating group mean ± SD with individual mice represented as a single datapoint (**b** PBS and BnT-Pfizer $n = 5$, and RBD and RBD-MSA $n = 6$, or **c** $n = 5$, except RBD-MSA $n = 6$). **d** All mediastinal lymph nodes per group (PBS and BnT-Pfizer $n = 5$, and RBD and RBD-MSA $n = 6$) were merged and samples prepared in technical triplicates for flow cytometry. **b** One-way ANOVA with Tukey's multiple comparison and **c** Kruskal–Wallis test with Dunn's multiple comparison. **a** Created or **b** and **c** partially created in BioRender[86].

Moreover, with the low IgA responses measured at local mucosal sites after the intramuscular mRNA vaccination in mind (Fig. 3b), we compared intranasal and intramuscular delivery of RBD-QMP in the presence of CpG using the Tg32-hFc mice (Fig. 4e) as described above (Fig. 3a). Intramuscular delivery resulted in undetectable levels of RBD-specific IgA in BALF. The IgG responses were also lower compared with those seen after intranasal vaccination, both locally and systemically (Fig. 4e).

To address whether the robust generation of antibodies after intranasal vaccination was related to the increased molecular weight of RBD-fused albumin compared with RBD alone, we aimed to design a fusion protein with a similar size without having the ability to engage FcRn. Specifically, we chose mouse transferrin (Tf) (74.9 kDa) as a carrier, which is known to be recycled by the Tf receptor and is present at mucosal surfaces[53,54]. The designed RBD-Tf protein (Supplementary Fig. 7a) was produced in the same system as RBD-fused albumin, which yielded a pure monomeric fraction after purification with expected molecular weight (Supplementary Fig. 7b). The RBD-Tf bound epitope-specific SARS-CoV-2 antibodies (Supplementary Fig. 7c) and human ACE2, as well as mouse Tf receptor 1 (Supplementary Fig. 7d and e). Next, we intranasally vaccinated Tg32-hFc mice with equimolar amounts of RBD-QMP and RBD-Tf (Fig. 3a). This resulted in low levels of RBD-specific antibodies systemically and mucosally in mice receiving RBD-Tf, compared with when using QMP albumin as a carrier (Fig. 4f), and poor blocking activity in the ACE2-RBD inhibition assay (Fig. 4g).

To explore the versatility of the vaccine strategy, we fused the ectodomain of hemagglutinin (HA) from influenza A H1N1 (A/Puerto Rico/8/1934; PR8), containing both the globular domain and stalk, to the engineered albumin variant QMP or mouse Tf (Supplementary Fig. 8a). The fusion proteins were well produced and purified monomeric fractions bound epitope specific monoclonal antibodies targeting the HA globular domain, as well as the receptors (Supplementary Fig. 8b–d). Upon intranasal vaccination of Tg32-hFc mice, only HA-QMP induced robust HA-specific IgG responses systemically and mucosally, as well as IgA responses locally (Supplementary Fig. 8e). Importantly, challenge with a lethal dose of influenza A H1N1 PR8 virus five weeks after initial vaccination resulted in 86% survival of mice vaccinated with HA-QMP, compared with 43% for the HA-Tf group (Supplementary Fig. 8f).

In line with these data, when HA was fused C-terminally to MSA or AQ, a MSA variant with two amino acid substitutions (K500A/H510Q) that reduce binding to mouse FcRn[33], the fusion proteins bound as expected to mouse FcRn as well as to epitope specific monoclonal antibodies targeting the HA globular domain (Supplementary fig. 9a–d). Intranasal vaccination (Supplementary Fig. 9a) induced higher systemic HA-specific IgG responses for MSA-HA than for AQ-HA (Supplementary Fig. 10a) in BALB/c mice, while very low to undetectable responses were found in mice given unfused HA or NaCl (Supplementary Fig. 10a). Both HA-specific IgG and IgA responses were detected in BALF samples collected at week six (Supplementary Fig. 10b), and challenge with a lethal dose of influenza A H1N1 PR8 virus eight weeks after initial vaccination resulted in complete survival of mice vaccinated with MSA-HA, compared to only 40% for the vaccine design with reduced capacity to engage FcRn (AQ-HA), while no mice survived in the group given unfused HA (Supplementary Fig. 10c). In a long-term memory study where mice were challenged with influenza A H1N1 PR8 four and a half months after initial vaccination with MSA-HA, complete protection was still observed (Supplementary Fig. 10d).

We then explored how direct conjugation of the adjuvant CpG to RBD-QMP would affect the antibody responses. This was done by taking advantage of a free cysteine at position 34 in domain I of HSA, distally located from its principal FcRn binding site[32,55]. The thiol of this cysteine was targeted for site-specific covalent conjugation of CpG

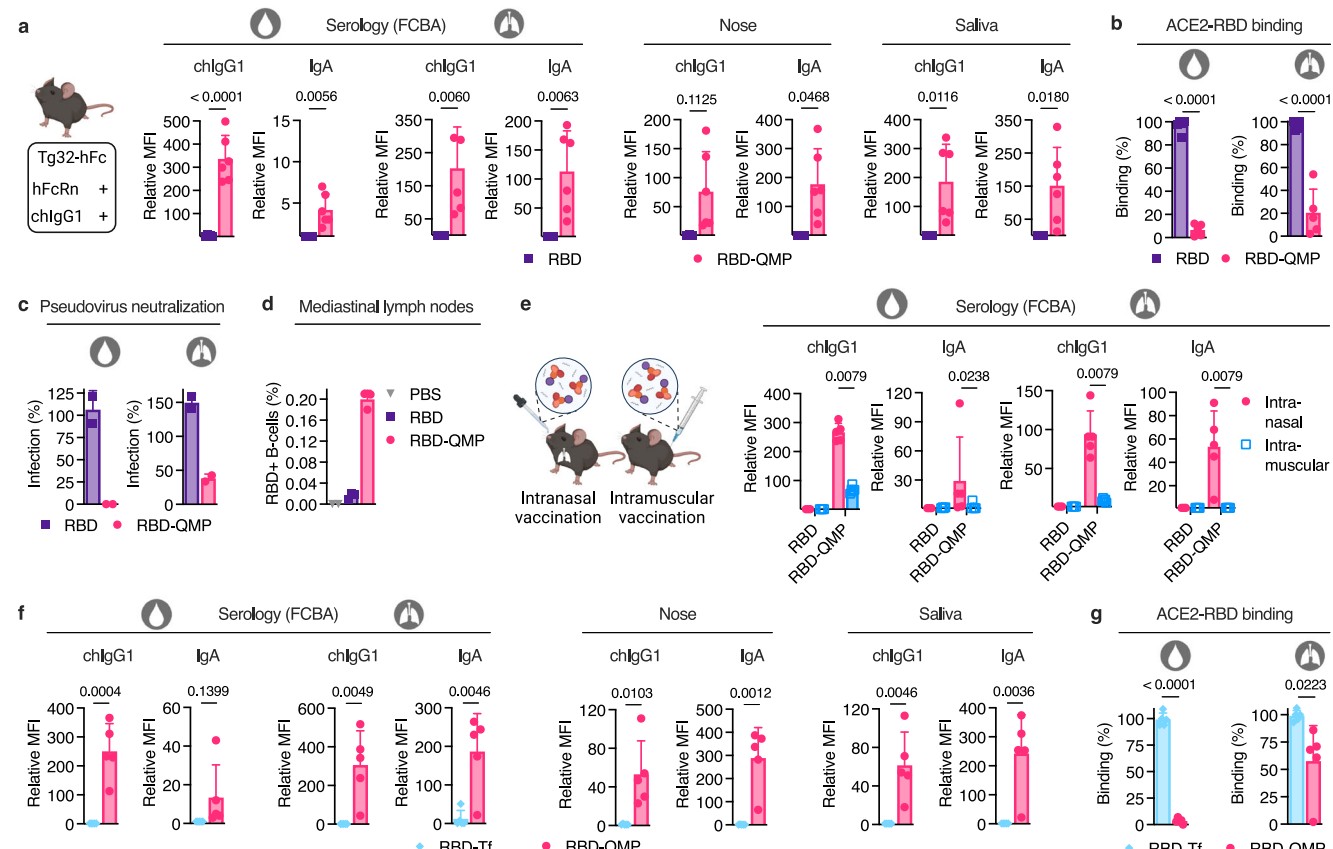

**Fig. 4 | Intranasal vaccination with RBD-fused engineered HSA in Tg32-hFc mice induces robust antigen-specific antibodies beyond that of intramuscular vaccination and of an alternative carrier. a** Illustration of Tg32-hFc mice expressing human FcRn and chimeric IgG1 with human Fc (chIgG1), and RBD-specific IgG and IgA in sera, BALF, nose and saliva of female mice at endpoint (prime dose: 6.2 μg RBD and 20.0 μg RBD-QMP in combination with 20 μg CpG), and the ability of the antibodies in sera and BALF to **b** inhibit human ACE2-RBD binding and **c** inhibit cellular infection by SARS-CoV-2 pseudovirus of the ancestral strain (Wuhan). **d** Percentage of antigen-specific B-cells in mediastinal lymph nodes. **e** Illustration of Tg32-hFc mice that were intranasally or intramuscularly vaccinated with RBD-QMP, and RBD-specific IgG and IgA in sera and BALF 5 weeks after initial vaccination of male mice (prime dose: 20.0 μg RBD-QMP in combination with 20 μg

CpG) **f** RBD-specific IgG and IgA in sera, BALF, nose and saliva of female and male Tg32-hFc mice at endpoint (prime dose: 20.0 μg RBD-QMP (2 F, 3 M) and 22.0 μg RBD-Tf (3 F, 2 M) in combination with 20 μg CpG), and **g** the ability of the antibodies in sera and BALF to inhibit human ACE2-RBD binding. All serology was detected using FCBA. Bars indicating group mean ± SD with individual mice represented as a single datapoint (**a** RBD $n = 5$ and RBD-QMP $n = 6$, except nose RBD $n = 3$, **b**, **e**, **f**, **g** $n = 5$, except **e** RBD $n = 3$ (intranasal) or $n = 4$ (intramuscular)). Pseudovirus neutralization **c** presented as technical duplicates of pooled biological samples per group (RBD $n = 5$ and RBD-QMP $n = 6$). **d** All mediastinal lymph nodes per group (PBS $n = 3$, RBD and RBD-QMP $n = 5$) were merged and samples prepared in technical triplicate (PBS as duplicate) for flow cytometry. **a**, **b**, **f**, **g** Two-tailed unpaired t-test, and **e** Two-tailed Mann–Whitney test. **a–c**, **e–g** Partially created in BioRender[86].

using maleimide chemistry (Fig. 5a). After successful conjugation and isolation (Supplementary Fig. 5b, c), binding of the resulting product to ACE2, epitope-specific SARS-CoV-2 antibodies, as well as human FcRn, was confirmed and shown to be similar to that of RBD-QMP (Supplementary Fig. 5d–f). Following intranasal vaccination of Tg32-hFc mice, the generated antibody responses in sera and at mucosal sites were similar between groups given CpG-RBD-QMP and RBD-QMP with CpG in solution (Fig. 5b and Supplementary Fig. 11a–c). However, an increased population of RBD-specific B-cells was measured in the mediastinal lymph nodes for CpG-conjugated RBD-QMP (Fig. 5c). Only minor antigen-specific immune responses were detected when an albumin fusion was intranasally administered in the absence of CpG, and this was independent of the vaccine subunit used (Fig. 5b and Supplementary Fig. 11a–d).

Another key question was whether the use of albumin as a vaccine subunit carrier could lead to antibody responses against albumin. The human FcRn-expressing mice used produce endogenous MSA possessing a 72% amino acid sequence similarity with HSA (Supplementary Fig. 12). Thus, we measured the levels of HSA-specific antibodies in vaccinated Tg32-hFc mice. Both CpG-RBD-QMP and RBD-QMP with added CpG induced HSA-specific IgG to the same levels, but we

measured very low responses in the absence of CpG (Fig. 5d). To investigate this further, we used mice transgenic for human FcRn, and with the gene encoding MSA replaced with the gene encoding HSA (HSA/hFcRn)[56] (Fig. 5e). Intranasal vaccination with RBD-QMP in the presence of CpG induced robust systemic and mucosal anti-RBD antibody responses (Fig. 5f and Supplementary Fig. 13a–c), but no antibody responses against HSA (Fig. 5g). Thus, HSA, even when engineered for improved receptor engagement, is not immunogenic in mice that constitutively express HSA. In line with this, BALB/c mice vaccinated with antigen-fused MSA in the presence of CpG did not raise antibody responses against MSA (Supplementary Fig. 14).

As a series of SARS-CoV-2 variants have emerged since the start of the pandemic, the amino acid composition of RBD has changed, which has caused concern regarding the protective effect of the first generation of vaccines[1,57]. Therefore, we studied binding of the antibodies induced upon intranasal vaccination with the Wuhan RBD to recombinant RBD proteins derived from B.1.1.7 (Alpha), B.1.351 (Beta), P.1 (Gamma), B.1.617.2 (Delta), B.1.427 (Epsilon), C.36 and BA.1, BA.2, BA.5 (Omicron) (Supplementary Fig. 15). The results demonstrated that sera, as well as locally and distally collected mucosal samples, from both vaccinated BALB/c mice and Tg32-hFc mice contained IgG and

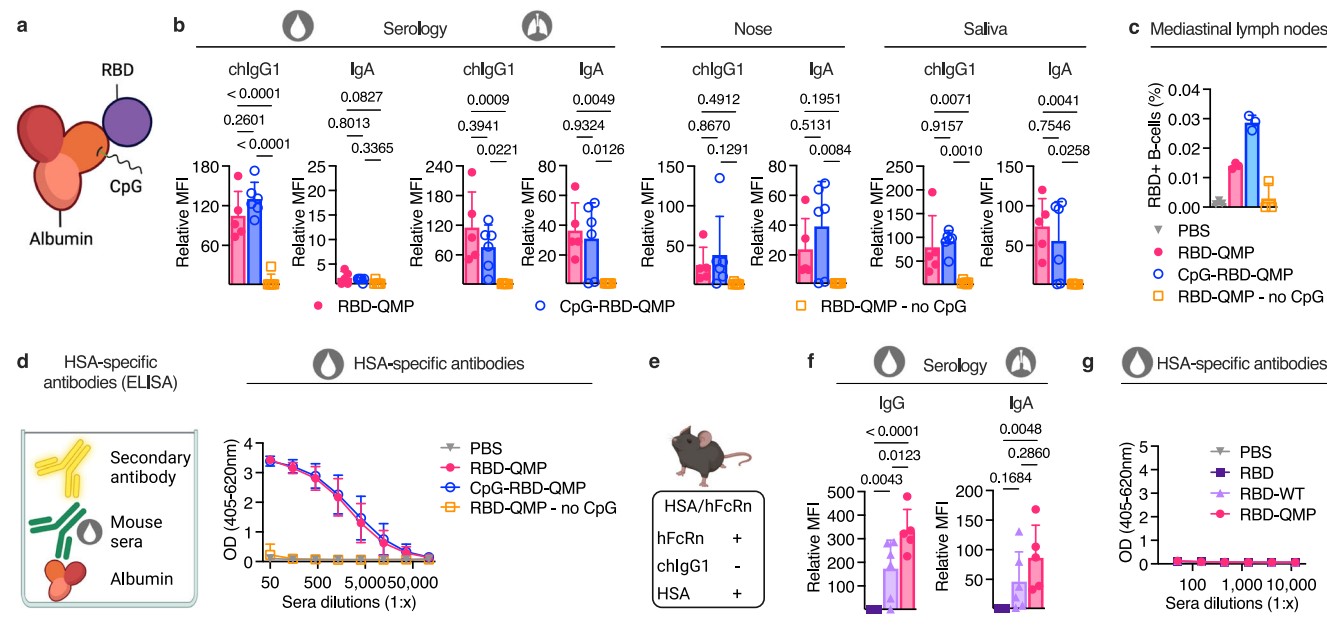

**Fig. 5 | Intranasal vaccination with RBD-fused engineered HSA induces antibodies solely toward the subunit vaccine antigen. a** Illustration of the RBD-fused albumin with CpG site-specifically conjugated to cysteine 34 (C34) in domain I of albumin, distal from the principal FcRn binding site. **b** RBD-specific IgG and IgA in sera, BALF, nose, and saliva at endpoint after intranasal vaccination of female Tg32-hFc mice (prime dose: 20.0 µg RBD-QMP with or without 20 µg CpG, 21.4 µg CpG-RBD-QMP, or PBS). **c** Percentage antigen-specific B-cells in mediastinal lymph nodes. **d** Illustration of ELISA for detection of antibodies against human albumin (HSA) after vaccination, and HSA-specific antibodies generated at endpoint after intranasal prime and boost with QMP-containing albumin detected using human IgG. **e** Illustration of the HSA/hFcRn mouse model expressing human FcRn and HSA.

Immune responses at endpoint after intranasal vaccination of female mice (prime dose: 6.2 µg RBD, 20.0 µg RBD-WT and 20.0 µg RBD-QMP in combination with 20 µg CpG) displaying **f** RBD-specific IgG and IgA antibodies in sera and BALF, respectively, and **g** HSA-specific mouse IgG antibodies in sera. Serology was detected using FCBA (**b** and **f**) or ELISA (**d** and **g**). Bars indicating group mean ± SD with individual mice represented as a single datapoint (except **d**) and curve plots presented as biological mean per group ± SD. **b**, **d**, **f**, **g** $n = 6$, except RBD-QMP $n = 5$. **c** All mediastinal lymph nodes per group $n = 6$, except RBD-QMP $n = 5$ were merged and samples prepared in technical triplicate for flow cytometry. **b** and **f** One-way ANOVA with Tukey's multiple comparison. **a** and **e** Created or **b**, **d**, **f**, **g** partially created in BioRender[86].

IgA antibodies capable of binding to all RBD variants (Supplementary Fig. 16a and d). Moreover, the antibodies generated could inhibit the human ACE2-RBD interactions, though the inhibition was reduced for the more recent mutated variants, such as omicron (Supplementary Fig. 16b and e), in line with the ability to block cellular infection from pseudoviruses displaying SARS-CoV-2 Spike variants (Supplementary Fig. 16c and f). Thus, intranasal vaccination of RBD-fused albumin induced antibody responses that could inhibit a range of SARS-CoV-2 variants.

In summary, we report on a simple vaccine design where an antigen of interest is directly fused to albumin targeting mucosal FcRn, which was explored in four different mouse strains (BALB/c, K18-hACE2, Tg32-hFc, HSA/hFcRn). The vaccine strategy induced antigen-specific IgG responses that were at least as robust as those obtained after intramuscular mRNA vaccination, while vaccination with antigen alone or as a fusion design to Tf, an alternative carrier of similar size to albumin, induced very low levels of antibody responses. Importantly, intranasal vaccination with the antigen given as an albumin fusion also yielded mucosal antigen-specific IgA responses throughout the airways. However, species differences exist between mice and men regarding IgA. For instance, humans have systemic monomeric IgA that engages the Fcα receptor on immune cells, and dimeric IgA that engages the pIgR at mucosal epithelial cell surfaces. In contrast, mice mainly express dimeric IgA in circulation and lack an Fcα receptor orthologue. They do, however, express the pIgR that can transport higher-order IgA oligomers across mucosal barriers followed by secretion[58–60]. In our case, we found that IgA at mucosal surfaces displayed oligomeric forms, particularly in the nose and saliva (Supplementary Fig. 17).

To our knowledge, CanSinoBIO (Convidecia Air/Ad5-nCOV-IH) and Bharat Biotech's (iNCOVACC/BBV154) nasal COVID-19 vaccines

were domestically approved based on antibodies detected in sera showing neutralizing capacity[23,61–65]. Notably, Ad5-nCOV-IH has also been shown to induce salivary IgA responses 14 days after vaccination, similar to that measured after intramuscular vaccination with the same vaccine[66]. Interestingly, the nasal SARS-CoV-2 adenovirus vaccine from AstraZeneca-Oxford University (Vaxzevria/Covishield/AZD1222) was discontinued after the phase I human trial (NCT04816019) because of low antigen-specific antibody immunity[65,67]. In fact, far from all participants generated detectable mucosal antibodies, while the systemic responses were weaker than those observed after intramuscular vaccination. The data from pre-clinical studies showed detectable levels of IgA in the nose and BALF of non-human primates, as well as reduced viral load in the lungs of hamsters and non-human primates[65,67,68]. IgA in BALF and nasal swabs from non-human primates have been reported after intramuscular mRNA vaccination with Spikevax (mRNA-1273, Moderna)[12], though a study that included triple-vaccinated Swedish healthcare workers did not find increased mucosal IgA responses after intramuscular vaccination alone, only when in combination with natural infection, and there were no difference between vaccine formulas (AstraZeneca-Oxford University, BioNTech-Pfizer or Moderna)[8]. The necessity of breakthrough infection for robust levels of IgA in the upper respiratory tract was confirmed in a study of American healthcare workers[69]. In accordance with this, we did not detect reasonable amounts of IgA in the nose, saliva or BALF of mice given the BioNTech-Pfizer mRNA vaccine intramuscularly, but instead IgG at the same mucosal sites, underlying the ability FcRn has to transport IgG, but not IgA, across mucosal barriers.

More recently, mucosally delivered nanoparticle- and adenovirus-based SARS-CoV-2 vaccines have been explored in mice, hamsters, and non-human primates and shown to raise detectable levels of IgG in sera

and BALF and IgA in BALF, which protected against infection and reduced transmission[19–21]. In addition, intranasal vaccination with a SARS-CoV-2 spike protein fused to a human IgG1 Fc has been demonstrated to induce similar local and systemic antigen-specific IgG levels as intramuscular delivery, while IgA was only detected locally upon intranasal vaccination of WT mice[36], in line with our study. However, this study does not fully appreciate the distinct cross-species differences where the human IgG Fc will bind strongly to mouse FcRn than the human form[45,50]. When HA was fused to mouse IgG2a Fc and tested in WT mice, both IgG and IgA responses were measured in nasal wash[37]. While both IgG Fc and albumin interact with FcRn, there are major differences in ligand binding properties between mice and humans that must be considered in a translational perspective when exploring concepts built on these ligands as scaffolds. In our study, we have fully controlled this complexity using both MSA and HSA combined with four different mouse strains that take into consideration different aspects of FcRn biology. As such, we provide protocols for how vaccine studies could be explored with suitable model systems. Our rationale for using albumin as a vaccine carrier instead of the IgG Fc is based on our discovery that more HSA than human IgG is taken up across mucosal surfaces in human FcRn transgenic mice, which could be further enhanced by HSA engineering for improved human FcRn engagement[33]. How the two ligands as vaccine fusion carriers may differ regarding transport and induction of immune responses should be addressed by, for instance the experimental setups described by us.

Thus, there is a need to explore innovative vaccine strategies aiming for the induction of both protective systemic and mucosal immunity, where study designs are carefully planned pre-clinically by taking cross-species differences into consideration. The study reported here is an example, where we take advantage of the complex biology of FcRn. The work should inspire further design of HSA-based subunit vaccines that offer the advantage of convenient, needle-free delivery with high patient compliance, by use of nasal spray or inhalator.

## Methods

### Production of soluble human and mouse FcRn
Soluble truncated human FcRn and mouse FcRn were produced using a baculovirus expression vector system[70–72]. Briefly, semi-adherent High Five cells (Cat. B85502, Invitrogen) were grown until confluency in flat culture flasks with Express Five SFM Medium (Cat. 10486, Gibco) at 27 °C before suspension and adjustment to $1 \times 10^6$ cells/mL and transfer to Erlenmeyer flasks, which were then placed in an orbital shaker kept at 27 °C and 160 rpm. Next, the cells were transfected using a viral stock of baculovirus containing a vector encoding a soluble, truncated form of either mouse or human FcRn[70,72]. The viral stocks were kindly gifted by Prof. Sally Ward (University of Southampton, UK). Cells were kept under orbital shaking for 72 h, and at 23-24 °C if transfected for hFcRn or 27–28 °C if transfected for mFcRn. After 72 h, supernatant was harvested and filtrated though a 0.2 μm filter before affinity chromatography on HisTrap, as described below.

### Vector design and production of HA and HA fusion proteins
The cDNA fragment encoding aa 18-541 of HA from influenza A H1N1 (A/Puerto Rico/8/1934; PR8) was used to design mouse serum albumin (MSA) with HA fused to its C-terminal end. Briefly, HA with an upstream cDNA sequence encoding a GGSGGSGGSGGSGG-linker was cloned into a naked pcDNA3 (Cat. V79020, Invitrogen) vector (pcDNA3-HA), using the restriction sites XhoI and ApaI. A silence mutation of the HA-encoding cDNA fragment was done to remove an additional XhoI restriction site before subcloning. Next, the cDNA fragment encoding MSA (wild-type or K500A/H510Q (AQ)) was cloned on the restriction sites HindIII and XhoI (pcDNA3-MSA-HA) for the production of MSA-HA or AQ-HA. Two HindIII restriction sites were removed in the MSA encoding cDNA sequence by silence mutation to enable subcloning of

fragments encoding MSA. A pLNOH2-vector with the cDNA for HA with a C-terminal 6xHis-tag was used for the production of His-tagged HA. The vectors were transiently transfected into adherent HEK293E cells (Cat. CRL-1573, ATCC)[73]. Briefly, cells were kept at 37 °C and 5% $CO_2$ and transfected using Lipofectamine 2000 Transfection reagent (Cat. 11668500, Invitrogen) according to the manufacturer's protocol. Supernatant was harvested every other day for two weeks.

The HA-Tf and HA-QMP fusions were made by synthesizing and cloning (GenScript) the cDNA encoding 18-519 of HA from influenza A H1N1 (A/Puerto Rico/8/1934; PR8) followed by either full-length mouse Tf and a 6xHis-tag or full-length human serum albumin (HSA) containing the amino acid substitutions E505Q/T527M/K573P (QMP)[33] into pFUSE2ss-CLIg-hk (Cat. pfuse2ss-hclk, InvivoGen) using the restriction sites EcoRI and NheI. The resulting vectors were transiently transfected into Expi293F cells (Cat. A14527, Gibco) using Expifectamine 293 Transfection Kit (Cat. A14525, Gibco) according to the manufacturer's instructions.

### Vector design and production of RBD and RBD fusion proteins
cDNA encoding 6xHis-tagged RBD of ancestral SARS-CoV-2 (Wuhan) was expressed in the plasmid pCAGGS[74], whereas cDNA encoding 10xHis- and Avi-tagged RBD were synthesized and cloned (GenScript) into pFUSE2ss-CLIg-hk using the restriction sites EcoRI and NheI. Mutagenesis was performed in the vector of 6xHis-RBD to generate a panel of SARS-CoV-2 variants (GenScript). To generate RBD fusion proteins, the RBD (Wuhan) was N-terminally fused to MSA, HSA (WT or QMP) or Tf. RBD fused to Tf was designed with a C-terminal 6xHis-tag. The vectors were transiently transfected into Expi293F cells using the ExpiFectamine 293 Transfection Kit according to the manufacturer's instructions. 6xHis-tagged RBD was used for in vitro applications, whereas 10xHis- and Avi-tagged RBD was used for in vivo vaccination.

### Protein purification and verification
Purification of MSA-HA variants and His-tagged HA was conducted using a CNBr-activated Sepharose 4 Fast Flow (Cat. C5338, Invitrogen) coupled to anti-HA mAb H36-4-52[75], packed in a 5 mL column (Repligen) coupled to a BioLogic workstation and recorder (BIO-RAD). Briefly, approximately 10 column volumes (CV) of 1x Phosphate Buffered Saline (PBS) (Cat. D8537, Sigma-Aldrich)/0.05% sodium azide (Cat. S2002, Sigma-Aldrich) (pH 7.2) was used to pre-equilibrate the column before supernatant (sterile filtrated with a 0.22 μm vacuum filter (Corning) containing 0.05% sodium azide) was applied with a flow rate of 1-2 mL/min. Subsequently, equilibration was done using 10-30 CV of 1xPBS, and bound MSA-HA variants or His-tagged HA were eluted with 5-10 CV of 0.1 M glycine-HCl (Cat. G7126, Sigma-Aldrich) (pH 2.7). Similarly, the His-tagged FcRn receptors, RBD-variants and transferrin fusions were purified using HisTrap HP 5 mL columns (Cat. 17524801, Cytiva). The column was pre-equilibrated with 1xPBS, and for equilibration 1xPBS was followed by 25 mM imidazole (Cat. 56748 Sigma-Aldrich) in PBS (pH 7.2) was used for equilibration, before elution with 250 mM imidazole in 1xPBS (pH 7.3). HSA fusions were purified using columns packed with CaptureSelect Human Albumin Affinity Matrix (Cat. 1912970, Thermo Scientific) by Repligen and were pre-equilibrated and equilibrated with 1xPBS before elution with buffer consisting of 2 M $MgCl_2$ (Cat. 63064, Sigma-Aldrich) and 20 mM Tris (Cat. T6066, Sigma-Aldrich) (pH 7.0). The RBD-MSA-fusions were purified using columns packed with AlbuPure selective affinity chromatography adsorbent (Cat. 3151, Astrea), pre-equilibrated and washed with 50 mM sodium phosphate (Cat. 342483, Sigma-Aldrich) (pH 6-8) and eluted with buffer containing 50 mM ammonium acetate (Cat. 09689, Sigma-Aldrich) and 20 mM sodium octanoate (Cat. C5038, Sigma-Aldrich) (pH 7.0). Eluted fractions were collected, upconcentrated and buffer-changed to 1xPBS using Amicon Ultra Centrifugal Filter Units (Merck Millipore) with appropriate cutoff (10-100 K) and volume (0.5 mL, 4 mL or 15 mL). To ensure monomeric

fractions of purified proteins, size exclusion chromatography was performed using a Superdex 200 Increase 10/300 GL Column (Cat. 28-9909-44, Cytiva) coupled to an ÄKTA Avant 150 Chromatography System (Cytiva) run using Unicorn Software (Cytiva) prior to up-concentration by Amicon Ultra Centrifugal Filter Units.

Protein concentrations were determined using a DS-11+ (M/C) Spectrophotometer (DeNovix) programmed with the protein's extinction coefficients and molecular weights, while protein purity was assessed by non-reduced SDS-PAGE on a Bolt 12% Bis-Tris Plus Gel (Cat. NW00125, Invitrogen) and compared to Spectra Multicolor Broad Range Protein Ladder (Cat. 26623, Thermo Scientific). Briefly, 2 µg of the protein samples were prepared in 4X Bolt LDS Sample Buffer (Cat. B0007, Invitrogen) and Milli-Q water, according to the manufacturer's instructions. The gel was run in 1X Bolt MES SDS Running Buffer (Cat. B0002, Invitrogen) for 22 min at 200 V with a PowerPac HC power supply (Bio-Rad) and stained with Bio-Safe Coomassie Stain (Cat. 1610786, Bio-Rad).

### Direct conjugation of adjuvant to albumin fusion

A purified fraction of RBD-QMP was directly conjugated to the adjuvant. First, CpG-linkers were made. In brief, 5' amine-modified CpG 1826 (Ordered from Integrated DNA Technologies (/5AmMC6/TCCATGACGTTCCTGACGTT), all PS backbone) was conjugated to an NHS-ester-modified linker by mixing DMSO (Cat. 34869, Sigma-Aldrich), 1 mM CpG 1826, 0.1 M 4-(2-hydroxyethyl)-1-piperazineethanesulfonic acid (HEPES) (Cat. H4034, Sigma-Aldrich), and 100 mM Maleimide-(PEG)8-NHS-ester (SM(PEG)8) linker (Cat. 746207, Sigma-Aldrich) at a volume ratio of 3:2:10:1, respectively. After overnight incubation at room temperature (RT) with 650 rpm orbital shaking, excess linker was removed by ethanol precipitation, which entailed mixing CpG, absolute ethanol (EtOH) (Cat. 20821.310, VWR), and 3 M sodium acetate (Cat. S2889, Sigma-Aldrich) in a 5:31:4 volume ratio and incubate at −18 °C for 3–4 h followed by centrifugation at 17,000 x $g$ for 45 min. The pellet was then washed with EtOH and centrifuged again at 17,000 x $g$ for 30 min. The supernatant was aspirated, and the pellet was dried for 10 min and subsequently dissolved in nuclease-free water (Cat. AM9937, Invitrogen).

Subsequently, the CpG-linkers were conjugated to RBD-QMP. RBD-QMP was mixed with a 1.3 molar excess of CpG-maleimide in 0.1 M HEPES (pH 7) and incubated overnight at RT with 650 rpm orbital shaking.

The CpG-RBD-QMP was purified with HPLC using IEX chromatography with a Mono Q 5/50 GL anion exchange column (Cytiva) running on Chromeleon (Thermo Scientific), and included an UltiMate AFC-3000 automated fraction collector connected to an LPG-3400RS pump and a VWD-3400RS detector (Thermo Scientific). The sample was filtered through a 0.2 µm polypropylene filter (Kinesis) before injection in the HPLC, and purification was run at 0.3 mL/min using the following program: 5 min buffer A (20 mM Tris and 10 mM NaCl (pH 7.6)), a 70 min gradient to buffer B (20 mM Tris and 3 M NaCl (pH 7.6)), 5 min of buffer B, a 5 min gradient to buffer A and 5 min of buffer A. Absorbance was measured at 224 and 260 nm. The collected fractions (min 29 to 35) were pooled, desalted and up-concentrated using 10 kDa cut-off Amicon Ultra Centrifugal Filter Units.

Protein purity after direct conjugation was assessed using native PAGE. A 10% native PAGE gel was made by mixing 3.4 mL ProtoGel (30%) (Cat. EC-890, National Diagnostics), 5.6 mL Milli-Q water, 1 mL 10X UltraPure TBE Buffer (Cat. 15581-044, Invitrogen), 100 µL 10% Ammonium Persulfate (Cat. 17874, Thermo Scientific), and 10 µL TEMED (Cat. 17919, Thermo Scientific). The mixture was poured into 1 mm cassettes and let to polymerize for 45 min with a 10-well comb (Invitrogen). 10 pmol sample was mixed with 10% glycerol (Cat. 24388.295, VWR) and 1 g/L Orange G (Cat. O3756, Sigma-Aldrich), loaded on a gel, and run in 1X UltraPure TBE buffer at 150 V for ca 75 min at RT with an EPS 601 electrophoresis power supply (Amersham

Biosciences). The SYBR Gold (Cat. S11494, Invitrogen) staining was performed by submerging the gel in 1X SYBR Gold in Milli-Q water for 15 min. After SYBR Gold image acquisition, the gel was stained with Coomassie by submerging the gel in 0.3 g/L Coomassie Brilliant Blue R (Cat. B7920, Sigma-Aldrich), 4.5% methanol (Cat. 34860, Sigma-Aldrich) and 1% acetic acid (Cat. 1.01830, Supelco) for 15 min with orbital shaking. The gel destaining was performed in 10% methanol and 7.5% acetic acid overnight. Imaging of SYBR Gold and Coomassie stained gels was performed using Image Lab (Bio-Rad) on a Gel Doc EZ Imager (Bio-Rad).

### Human endothelial cell-based recycling assay (HERA)

HERA was performed on RBD-HSA fusions according to a published protocol[76,77]. Briefly, the human microvascular endothelial cell line-1 (HMEC-1) stably expressing HA-human FcRn-EGFP (HMEC-1-hFcRn, kind gift from Dr. Wayne I. Lencer (Boston Children's Hospital, Harvard Medical School and Harvard Digestive Diseases Center, USA)) ($7.5 \times 10^4$ viable cells/well) were seeded into Costar 48-well TC-treated Multiple Well Plates (Corning) and cultured in cell culture medium of MCDB 131 Medium (Cat. 10372019, Gibco) supplemented with 10% heat-inactivated fetal bovine serum (FBS) (Cat. F7524, Sigma-Aldrich), 1xPenicillin-Streptomycin (Cat. P4458, Gibco), 2 mM L-Glutamine (Cat. 25030081, Gibco), 10 ng/mL Recombinant Mouse Epidermal Growth Factor (Cat. PMG8041, Gibco), 1 µg/mL hydrocortisone (Sigma-Aldrich), 100 µg/mL Geneticin Selective Antibiotic (Cat. 10131035, Gibco), and 5 µg/mL Blasticidin S HCl (Cat. A1113903, Gibco) for 24 h at 37 °C in a humidified atmosphere of 5% CO2. The cells were then washed three times and starved in Hank's Balanced Salt Solution (HBSS) (Cat. 14025100, Gibco) for 1 h. After starvation, 800 nM of RBD-HSA fusions diluted in 125 µL of HBSS was added and incubated for 3 h followed by washing five times with cold HBSS. Subsequently, warm assay medium (complete cell culture medium without FBS, Blasticidin S HCl and Geneticin, but supplemented with 1xEagle's Minimum Essential Medium Non-Essential Amino Acids Solution (Cat. 11140050, Gibco)) was added for 3 h. The assay media were collected to determine the recycled amounts of fusions by ELISA, as described below.

### Transwell assay

A previously described transwell assay[29] was adapted to the Madin-Darby Canine Kidney II cell line that overexpresses human FcRn (MDCKII-hFcRn), and received as a kind gift from Dr. Alexander Haas and Jens Fischer (pRED Roche Innovation Center Munich, Germany). Briefly, 12 mm Transwell-COL Collagen-coated 0.4 µm Pore PTFE Membrane Inserts (Cat. 10457031, Corning) in Costar 12-well TC-treated Multiple Well Plates (Cat. 07-200-83, Corning) were initially equilibrated by adding complete cell culture medium (Dulbecco's Modified Eagle Medium (DMEM) with GlutaMAX Supplement (Cat. 31966, Gibco), added 10% FBS and 300 µg/mL Geneticin Selective Antibiotic), and incubating overnight. MDCKII-hFcRn cells ($1.25 \times 10^6$ viable cells/well) were then seeded into the inserts and incubated for 24–26 h at 37 °C in a humidified atmosphere of 5% CO2. The cells were then washed three times with pre-warmed HBSS, before transepithelial electrical resistance (($600$-$1100$) $\Omega\text{xcm}^2$) was measured using a Millicell ERS-2 Voltohmmeter (Millipore). Then, the cells were starved for 1 h in HBSS. Subsequently, 800 nM of RBD-HSA fusions diluted in 200 µL of HBSS was added to the transwell inserts (apical side) followed by sampling of 900 µL of incubation solution (HBSS) from the wells (basolateral side) after 4 h. The transcytosed amounts of the RBD-HSA fusions were determined by ELISA, as described below.

### In vivo vaccine studies

All mice included in this study were housed under controlled environments including ventilated cages with regularly change of bedding

and nesting material, ad libitum access to water and food, light/dark cycle of 12 h and regulated ambient temperature (21 °C ± 2 °C) and relative humidity (30-70%). BALB/c mice (BALB/cAnNRj, strain #0003 Janvier-Labs) (female, 6-8 weeks old; 5-12 mice per group (HA, AQ-HA and MSA-HA), or female, 8-9 weeks; 5 (PBS and BioNTech-Pfizer) or 6 (RBD and RBD-MSA) mice per group), Tg32-hFc mice (B6.Cg-Tg(FCGRT)32Dcr *Fcgrt*^tm1Dcr *Ighg1*^em2(IGHG1)Mvw/MvwJ, strain #029686, The Jackson Laboratory) (female and male, 6-13 or 31 weeks; 5-7 mice per group (except RBD group in intranasal vs intramuscular experiment - 3 and 4 mice, respectively) and HSA/hFcRn mice (C57BL/6N-*Fcrgt*^tm1.1(FCRGT)Geno:*Alb*^tm1.1(ALB)Geno, genOway) (female, 13-14 weeks, 6 mice per group) were used for SARS-CoV-2 and Influenza HA vaccine studies.

The mice were anesthetized by intraperitoneally administering 5–7.5 mL/kg of ZRF cocktail containing 250 mg/mL of Zoletil Forte, 20 mg/mL of Rompun, and 50 μg/mL of Fentanyl in saline solution (Section for Comparative Medicine, Rikshospitalet, Oslo University Hospital). After sedation, 20 μL of vaccination solutions containing 0.22 nM 10xHis-RBD (6.2 μg), RBD-MSA (19.9 μg), RBD-HSA (20.0 μg), or RBD-Tf (22.0 μg) fusion, or 0.16 nM His-tagged HA (13.1 μg), MSA-HA (27.5 μg), AQ-HA (27.5 μg), HA-QMP (26.9 μg) or HA-Tf (28.9 μg) mixed with 20 μg CpG ODN 1826 VacciGrade (Cat. Vac-1826-1, InvivoGen), or 0.22 nM RBD-QMP (20.0 μg) or CpG-RBD-QMP (21.4 μg), or 1xPBS or saline (NaCl) alone were intranasally administered to each mouse (10 μL/nostril). For intramuscular delivery in the quadriceps muscle, 0.22 nM RBD-QMP (20.0 μg) with 20 μg CpG, 3 μg of Comirnaty (BioNTech-Pfizer) or 1xPBS were given (15 μL/leg). A subsequent dose with 10% of the RBD, RBD-albumin, HA or MSA-HA fusion (0.022 nM) mixed with 20 μg CpG, 0.3 μg of Comirnaty or PBS/NaCl was given 3–4 weeks after the initial dose.

Blood samples were collected from the saphenous vein. For mice vaccinated with an HA-fusion, blood was collected at weeks 1, 3, 6 and 7.5 after vaccination. For mice vaccinated with an RBD-fusion, blood was collected at weeks 1, 2, 4 and/or 5 after vaccination. Sera were isolated by centrifugation at 17,000 x *g* for 10-25 min at 4 °C. In addition, mucosal flushes and bronchioalveolar lavage fluid (BALF) were collected at endpoint for selected experiments. All sites were flushed with sterile PBS, and except BALF all samples were directly transferred to protease inhibitor to a final concentration of x1-x2. The following volume of PBS was used: Nose – 15 μL in each nostril, saliva – 25 μL for each check with 3 aspirations per side, vaginal and rectal – 25 μL x 2 with 3 aspirations, BALF – 1000 μL. Samples with protease inhibitors were centrifuged at 17,000 x *g* for 10–25 min at 4 °C to remove debris. All samples were stored at −20 °C until analyses of antigen-specific immune responses, using ELISA or FCBA as described below. At endpoint mediastinal lymph nodes (also inguinal for PBS groups) were harvested for flow cytometry analyses. The animal experiments were carried out at the Department for Comparative Medicine, Oslo University Hospital (KPM Rikshospitalet; Oslo, Norway), with approval from the Norwegian Food Safety Authority, and performed in accordance with the Guide for the Care and Use of Laboratory Animals of the Norwegian National Institute of Health. The animal facility is classified as specific pathogen-free as defined by Federation of European Laboratory Animal Science Associations. Experimental and control animals were bred in the same facility and co-housed.

Vaccination of K18-hACE2 mice (B6.Cg-Tg(K18-ACE2)2Prlmn/J, strain #034860; The Jackson Laboratory) female, 10–12 weeks; 10 (PBS) or 12 (RBD and RBD-MSA) mice per group) were performed at the Association for the Assessment and Accreditation of Laboratory Animal Care (AAALAC) accredited The Scripps Research Institute (TSRI; La Jolla, CA, USA), and the experiment was approved by The Institutional Animal Care and Use Committee (IACUC) at TSRI. Vaccine mixtures and administration methods were as above, except anesthesia was achieved using inhaled isoflurane and blood samples were collected

through retroorbital bleed at days 14 and 28. Experimental and control animals were bred in the same facility and co-housed.

## Live virus challenge

The 5x Lethal Dose 50 (5xLD50) of mouse-adapted Influenza A H1N1 A/PR/8/34 virus (Cat. VR-95, ATCC) for BALB/c mice and Tg32-hFc mice were determined according to the Reed and Muench method[78,79]. Vaccinated mice were transferred to the BSL2 approved facility at KPM Rikshospitalet prior to challenge with a deadly dose (5xLD50) of PR8 either 5 weeks, 8 weeks or 4.5 months after initial vaccination. Specifically, the mice were anesthetized and intranasally administered with the virus (10 μL/nostril). Weight loss was monitored daily or every second or third day after infection, in which the endpoint was set at 20% weight reduction. If the endpoint was reached prior to the end of the experiment, mice were terminated by cervical dislocation or using a CO$_2$ gas chamber. PR8 challenge was performed with approval from the Norwegian Food Safety Authority.

Vaccinated K18-hACE2 mice were transferred to the BSL3 approved vivarium at TSRI and challenged with 2x10$^4$ pfu SARS-COV-2 WA1/2020 (kind gift from Dr. John Teijaro (TSRI, La Jolla, CA, USA)) as above, anesthetized by isoflurane. The mice were monitored and weighed daily for seven days, after which they were euthanized and the lungs harvested for analyses. The SARS-CoV-2 challenge was approved by the IACUC at TSRI.

## Histopathology

The left lung lobes were rinsed with PBS and fixed with 4% formalin at endpoint. At The Unit for Research Support at the Department of Pathology, Oslo University Hospital the fixed lungs were embedded in paraffin and cut at 3 μm thick slices before they were manually stained with ready-to-use hematoxylin and eosin (Harris HTX Histolab 1 L, cat. 01800, Histolab). Blinded microscopic analysis was performed by an educated pathologist. Pictures of a selection of slides were taken with an Olympus BX53 microscope connected to a XC50 camera using cellSens Entry.

## Viral RNA quantification

Viral RNA was isolated from half of each right lung lobe, and subsequently amplified and quantified by reverse transcription qPCR. The lung tissue was homogenized in 1 mL TRIzol reagent (Cat. 15596026, Invitrogen) using a Bead Ruptor 12 (Omni International). After centrifugation at 12,000 x *g* for 10 min at 4 °C, 750 μL supernatant was added to 200 μL chloroform, thoroughly mixed and spun again. The RNA containing aqueous phase was mixed with 350 μL of 70% EtOH for a 1:1 ratio, loaded onto the spin columns of RNeasy Mini kit (Cat. 74104, Qiagen) and purified according to the manufacturer's instructions. RNA was eluted using 25 μL of UltraPure RNase-free water (Cat. 10977035, Invitrogen). After normalization, 1.2 μg RNA was mixed with 2.5 μL UltraPlex 1-Step ToughMix (Cat. 95166, QuantaBio) and 0.75 μL of Centers for Disease Control and Prevention's N1 (nucleocapsid) primer sets (forward, 5′-GACCCCAAAAT-CAGCGAAAT-3′; reverse, 5′-TCTGGTTACTGCCAGTTGAATCTG-3′) (Cat. 10006821 and 10006822, Integrated DNA Technologies) and a fluorescently labeled (FAM) probe (5′-FAM-ACCCCGCAT-TACGTTTGGTGGACC-BHQ1-3′) (Cat. 10006823, Integrated DNA Technologies) in a total reaction volume of 10 μL. A standard consisting of 1.5x10$^6$ PFU equivalents of SARS-CoV-2 RNA was serially diluted 10-fold and a no template control was included in each run. PCR was performed using CFX96 Real-Time PCR Detection Systems (Bio-Rad), running on CFX Manager (Bio-Rad).

## ELISA

All consecutive ELISAs were performed according to the following protocol, unless otherwise stated. Corning 96-well EIA/RIA Polystyrene High Bind Microplates (Corning) were coated with 100 μL/well of the

desired protein diluted in 1xPBS overnight at 4 °C or 2 h at RT. Wells were blocked with 200 μL/well of PBS containing 4% (w/v) skimmed milk powder (S) (Cat. 115363, Millipore) (PBS/S) for 1 h at RT before being washed with 250 μL/well of PBS supplemented with 0.05% (v/v) Tween 20 (Cat. P1379, Sigma-Aldrich) (PBS/T), three times consecutively. Next, 100 μL/well of samples or proteins (50 μL/well for sera or BALF) diluted in PBS/T supplemented with 4% (w/v) S (PBS/T/S) were added, incubated for 1–2 h at RT, and washed as above and 100 μL/well of a secondary antibody was added for 1 h at RT, followed by another wash cycle. Development of horseradish peroxidase (HRP)-conjugated antibodies was done by adding 100 μL/well 3,3′,5,5′-Tetramethylbenzidine (TMB) solution (Cat. CL07, Calbiochem), and the reaction was stopped by adding 100 μL/well of 1 M HCl. Assays using alkaline phosphatase (ALP)-conjugated antibodies were developed by adding 100 μL/well p-nitrophenyl phosphate substrate tablets (Cat. S0942, Sigma-Aldrich) dissolved in diethanolamine buffer (pH 9.8) (made in-house) to a final concentration of 1 mg/mL (pNpp). Absorbance measurements were performed using a Sunrise spectrophotometer (TECAN) at 405 nm or 450 nm for pNpp or TMB substrates, respectively, with a reference wavelength of 620 nm.

For detection of His-tagged HA and HA-fusions Influenza A H1N1 (A/Puerto Rico/8/1934) HA ELISA Pair Set (Cat. SEK11684, Sino Biological) was used.

Quantification of RBD-HSA fusions from HERA and Transwell assay was done in a two-way anti-albumin ELISA, detecting present albumin from serial dilution of a standard curve or samples (HERA, 1:2-1:8 and Transwell, 1:1-1:450) between a goat anti-HSA (Cat. A1151, Sigma-Aldrich) (1:2,000) and goat anti-HSA-ALP (Cat. A80-229AP, Bethyl Laboratories, Inc.) (1:4,000).

To determine binding between FcRn and albumin fusions, 10 μg/mL of His-tagged mouse FcRn or human FcRn (made in-house) diluted in PBS/S/T (pH 5.5) was added to plates coated with 8 μg/mL of a human IgG1 mutant (M252Y/S254T/T256E/H433K/N434F)[80] (made in-house) with specificity for 4-hydroxy-3-iodo-5-nitrophenylacetic acid, followed by addition of serial dilutions of albumin fusions (225.6 nM-0.004 nM) in PBS/S/T (pH 5.5). For MSA-HA and AQ-HA, 39.4 nM-0.48 nM was used. MSA-HA and AQ-HA were detected with an anti-HA mAb H36-4-52 from mouse[75] (2 μg/mL) (made in-house), pre-incubated with anti-mIgG(Fc)-ALP (Cat. A2429, Sigma-Aldrich) (1:3,000). Albumin fusions were detected with either anti-MSA-HRP (Cat. ab19195, Abcam) (1,5000) or anti-HSA-ALP as above.

To investigate binding between ACE2 and RBD-albumin fusions, plates were coated with 2 μg/mL of His-tagged recombinant human ACE2 (Cat. 10108-H08H, Sino Biological) before serial dilutions of RBD-MSA and RBD-HSA fusions (150 nM-0.023 nM) were added, followed by detection with anti-MSA-HRP or anti-HSA-ALP.

Binding between ACE2 and RBD-Tf was confirmed by coating plates with 35 μg/mL RBD-Tf diluted 4-fold (349.60-0.08 nM) to an end volume of 100 uL. After blocking, plates were incubated with 1 μg/mL of a recombinant ACE2-HSA fusion protein (made in-house)[49]. Finally, goat anti-HSA-ALP was used for detection as described above.

Binding to monoclonal antibodies with known epitope specificity was used as a measure for epitope availability on the fusions. In short, HA and HA fusion proteins were tested by adding 15 nM samples to plates coated with anti-His (Cat. ab18184, Abcam) (1:2000), anti-MSA (Cat. ab19194, Abcam) (1:1000), or anti-HSA. A 3-fold dilution series of the following monoclonal anti-HA with known antigenic site[81] were added (mAbs from supernatant starting at 1:30 dilution and purified mAbs starting at 10 μg/mL, performed with 8 dilution steps); Y8-2C6, H28-E23, H17-L2, H36-11, H18-S413 and H9-A15 and detected using a biotinylated anti-mouse kappa light chain (made in-house, clone 187) (1:1000) and SA-ALP (Cat. 7105-04, SouthernBiotech) (1:3000). RBD and RBD fusion proteins (except RBD-Tf) were tested by adding a 4-fold serial dilution from 4 μg/mL (equimolar to RBD-HSA; 43.7-

0.003 nM) to plates coated with 1 μg/mL of the monoclonal SARS-CoV-2 antibodies sotrovimab (GSK), cilgavimab or tixagevimab (AstraZeneca), before detection using anti-His-ALP (Cat. Ab49746, Abcam) (1:8000), anti-MSA-ALP or anti-HSA-ALP as above. For the RBD-Tf fusion, a 4-fold titration from 17.6 μg/mL (174.80−0.01 nM) was used for coating, followed by the monoclonal SARS-CoV-2 antibodies as above, before detection by anti-hIgG Fc specific-ALP (A9544, Sigma-Aldrich) (1:5000).

For the detection of RBD- or HA-specific antibodies from sera and BALF, titrated amounts of sera (diluted 1:50-1:109,350) or BALF (diluted 1:2-1:2,048) were added to plates coated with 1 μg/mL recombinant 6xHis-RBD (made in-house) or 0.5 μg/mL HA from Influenza A H1N1 (A/Puerto Rico/8/1934) (Cat. 11684-V08H, Sino Biological), while for detection of MSA- or HSA-specific antibodies from sera, titrated amounts (diluted 1:50-1:109,350) were added to plates coated with 0.5 μg/mL of recombinant MSA (Cat. A3559, Sigma-Aldrich) or HSA (made in-house). Anti-mIgG(Fc)-ALP or anti-hIgG(Fc)-ALP (1:5,000) were used to detect total IgG. For detection of IgG subclasses, anti-mIgG1-bi, anti-mIgG2a-bi or anti-mIgG2b-bi (Cat. 553500, 553502, 553393, BD Pharmingen), all pre-incubated with Streptavidin-ALP (1:3000) were used, while IgA was detected using anti-mIgA-ALP (Cat. 3865-9 A, Mabtech) (1:1,000). Data were presented as OD values, or as antibody titers which equal to the highest dilution factor for each mouse with a higher OD value than the background, where the background is the mean absorbance of mice given PBS/NaCl plus 5x the standard error of the mean of the same observations.

Serum samples were tested for their ability to inhibit human ACE2-RBD binding in ELISA. Plates were coated with 0.25 μg/mL of 6xHis-RBD. Serial dilutions of serum samples were added (1:30 dilution presented). Next, equimolar amounts of human ACE2 fused to HSA WT (1.364 μg/mL) and anti-HSA-ALP were used to determine the amount of RBD-bound ACE2.

## Surface plasmon resonance

A Biacore T200 instrument (Cytiva) was used with 1xHBS-P+ (Cat. BR100671, Cytiva) as running buffer. 10 μg/mL of recombinant His-tagged mouse transferrin receptor (TFR1/CD71) (Cat. 50741-M07H, Sino Biological) was diluted in Acetate 4.5 (Cat. BR100350, Cytiva) and immobilized with the Amine Coupling Kit (Cat. BR10050, Cytiva) on a Series S CM5 Sensor chip (Cat. 29149603, Cytiva) to reach about 350 RU. Triplicates of 8,000 nM of either RBD-Tf or HA-Tf were injected using 177 mM phosphate, 85 mM NaCl, 0.005% Tween 20, pH 5.5 (made in-house) as running buffer, and 1x PBS supplemented with 0.005% Tween 20, pH 7.4 (made in-house) as regeneration buffer between every run. All runs were performed at 25 °C using a 40 μL/min flow rate, 120 seconds association phase and 680 seconds dissociation phase, with a 30 seconds regeneration phase at 30 μL/min in between each run. The binding response from a blank control flow cell was subtracted from all response curves to correct for background.

## Quantification of RBD-specific antibodies in FCBA

Previously, a bead-based flow cytometric assay was adapted for the detection of antibodies against RBD from ancestral SARS-CoV-2 (Wuhan) as well as SARS-CoV-2 variants[41,42]. Bead-based arrays with content of virus proteins were incubated for 1 h or overnight at RT, for measurements of IgGs and IgA, respectively, in sera and BALF, or overnight for measurements in other mucosal flushes, diluted in an assay buffer consisting of PBS supplemented with 1% Tween 20 (PBT), 1% Bovine serum albumin (BSA) (Cat. A7906, Sigma-Aldrich), 10 μg/mL neutravidin (NA) (Cat. 31000, Thermo Scientific), 10 μg/mL D-biotin (Cat. 2031, Sigma-Aldrich) and 0.1% sodium azide. The dilutions were as follows: serum – 1:100, or 1:300 for IgG in Tg32-hFc from experiment comparing intranasal and intramuscular vaccination from Fig. 4, or 1:900 for IgG in BALB/c from Fig. 1 and Supplementary Fig. 2, BALF –

1:10, or 1:30 for IgG in BALB/c from Fig. 1 and Supplementary Fig. 2, and mucosal flush −1:4 (nose; BALB/c and HSA/hFcRn), 1:6 (nose; Tg2-hFc), 1:3 (saliva; BALB/c and Tg32-hFc, and rectal; Tg32-hFc), 1:2 (saliva; HSA/hFcRn, and vaginal; all) or 1:1.5 (rectal; BALB/c and HSA/hFcRn)). The beads were then washed three times with PBT to remove unbound immunoglobulins and labeled for 30 min at RT with 10 μL of R-Phycoerythrin (PE)-conjugated goat anti-mIgG (Cat. 31861, Invitrogen), goat anti-hIgG (Cat. 109-115-098, Jackson Immunoresearch), rat anti-mIgG1, IgG2a, IgG2b, IgG3 (Cat. 1144-09, 1155-09, 1186-09, 1191-09, SouthernBiotech) or digoxigenin (DIG)-conjugated rat anti-mIgA (Cat. 3865-3, Mabtech). Stocks were diluted 1:200, 1:200, 1:100 and 1:100, respectively, in PBT containing 1% BSA and 0.1% sodium azide. After incubation with anti-mIgA-DIG, beads were washed three times before labeling for 30 min with 30 μL mouse anti-DIG-PE (Cat. 200-002-156, Jackson Immunoresearch) (0.5 μg/mL). Following a final wash, the beads were resuspended in PBT containing 0.1% BSA, and run on Attune NxT Flow Cytometer (Invitrogen). Specific antibody binding was measured as the ratio of PE median fluorescence intensity (MFI) of antigen-coupled beads to beads coupled with NA only, referred to as relative MFI (rMFI). The anti-mIgA-DIG conjugate was prepared by first changing the buffer of the antibody to PBS without sodium azide using 100 K cutoff Amicon Ultra Centrifugal Filter Units. Then, 1 mg/mL anti-mIgA was mixed with 200 μg/mL DIG-N-hydroxysuccinimidyl-ester (DIG-NHS) (Cat. 55865. Sigma-Aldrich) and incubated overnight. Excess DIG-NHS was removed by centrifugation through Amicon Ultra Centrifugal Filter Units with 100 K cutoff twice, before the solution was passed over a G50-Sephadex column (Cat. 17004202, Cytiva) equilibrated with PBS. Anti-DIG was conjugated to PE using standard thiol-maleimide chemistry.

## FCBA for detection of neutralizing antibodies
A published method for measuring antibody-mediated inhibition of ACE2-RBD interactions was adapted to a microsphere format[42]. Briefly, an aliquot of the bead-based arrays used for antibody measurement (see above) were incubated for 1 h at RT with serum (diluted 1:100, or 1:300 for BALB/c from Fig. 1 and Supplementary Fig. 2) or BALF (diluted 1:10, or 1:30 for BALB/c from Fig. 1 and Supplementary Fig. 2) in assay buffer. The supernatant was removed, and the beads were incubated with 20 μL of an ACE2-HSA-DIG (0.8 μg/mL) for 30 min at RT. Following two washes with PBT, the beads were labeled with 30 μL anti-DIG-PE (0.5 μg/mL) for 30 min at RT, washed 4 times with PBT before resuspension in PBT containing 0.1% BSA and 10 μg/mL D-biotin, and run on Attune NxT Flow Cytometer (Invitrogen). Percentage inhibition of the ACE2-RBD interaction was calculated by measuring the ratio of PE MFI of antigen-coupled beads to beads coupled with NA only (rMFI), followed by normalization to relative MFI-values from PBS vaccinated mice, which was defined as 0% inhibition. The ACE2-HSA conjugate was prepared in-house as described[42,49]. To confer readability and direct comparison with the pseudovirus neutralization assays, no detection of neutralizing antibodies in FCBA (0% inhibition) was defined as 100% binding.

## Preparation of Spike (S) pseudotyped HIV-1 virions
Replication deficient SARS-CoV-2 pseudotyped HIV-1 virions were prepared as described previously[43]. Briefly, virions were produced in 293T WT (293T-WT) cells by transfection with 1 μg of the plasmid encoding SARS CoV-2 Spike protein (pCAGGS-Spike Δc19)[43], 1 μg pCRV Gag-Pol[82] and 1.5 μg GFP-encoding plasmid (CSGW)[83]. Viral supernatants were filtered through a 0.45 μm syringe filter at 48 and 72 h post-transfection and pelleted for 2 h at 28,000 x g. Pelleted virions were drained and then resuspended in DMEM.

## Spike pseudotyped neutralization with mouse sera or BALF
293T-hACE2-TMPRSS2 cells[43] were plated into 96-well plates at a density of 0.75 x 10³ cells per well and allowed to attach overnight.

20 μL pseudovirus was mixed with 2 μL dilutions of heat-inactivated mouse sera (made in-house) (1:100) or BALF (1:7 or 1:8.4) and incubated for 40 min at RT. 10 μL of this mixture was added to the cells. 72 h later, infection was detected through GFP expression by visualization on an Incucyte S3 live cell imaging system (Sartorius). The percentage of infection was quantified as GFP-positive cell area over total cell area. Relative infection was calculated as percentage virus infection in the presence of immune sera/BALF relative to virus only control.

Pseudovirus neutralization was performed on sera from mice[84]. The serum was diluted 1:10 in DMEM (Gibco) supplemented with 10% heat-inactivated FBS and serially diluted 3-fold to a total of 8 dilutions. 20 μL of diluted serum was mixed with 20 μL of lentivirus-based SARS-CoV-2 pseudovirus in a 384-well white tissue culture plate (Greiner Bio-One), and the plate was incubated for 60 min at 37 °C humidified atmosphere of 5% $CO_2$. Subsequently, 20 μL containing 5,000 HeLa-ACE2 cells were added per well, following 48 h of incubation, before the cell culture media was discarded. The cells were lysed with 25 μL lysis buffer (25 mM Gly-Gly pH 7.8 (Cat. G1002, Merck), 15 mM $MgSO_4$ (Cat. 83266, Sigma-Aldrich), 4 mM EGTA (Cat. E3889, Merck), 1% Triton X-100 (Cat. X100, Merck)) with added BrightGlo substrate (Cat. E2620, Promega) (1:10 dilution) and luciferase activity/luminescence was measured on a BioTek Synergy 2 microplate reader running Gen5. Percentage neutralization was calculated accordingly: 100 x (1 - (RLU of sample - Average RLU of CC) / (Average RLU of VC-Average RLU of CC)), RLU referring to relative light unit. Values were normalized so negative neutralization percentages were set to equal 0%.

## Flow cytometry on mediastinal lymph nodes
Collected mediastinal lymph nodes from mice were pooled according to treatment group and kept on ice or at 4 °C continuously from harvest to analysis and protected from light after addition of viability dye. Between steps, cells were centrifuged at 400 x g for 7 min at 4 °C before transfer to a 96-well plate, and 500 x g for 5 min at 4 °C after plating. Lymph nodes were mashed through a 70 μm cell strainer (VWR) to a single-cell suspension, before treatment with 3 mL RBC lysis buffer (150 mM $NH_4Cl$, 10 mM $Na_2CO$, 0.1 mM $Na_2EDTA$, made in-house) for 5 min. Cells were then washed with FACS buffer (2% FBS in PBS), and subsequently divided into triplicates of approximately 2 x 10⁶ cells. Next, the cells were stained with GhostDyeV510 (Cat. 13-0870-T100, Tonbo Bioscience) (1:400) for 20 min followed by blocking with 100 μL 25% normal rat serum (made in-house) in PBS for another 20 min.

The following surface markers were used: TCRβ-BB700 (Cat. 745846, BD Bioscience) (1:200), CD19-APC/Cy7 (Cat. 557655, BD Bioscience) (1:200), CD38-BUV395 (Cat. 740245, BD Bioscience) (1:200), CD45R(B220)-BUV805 (Cat. 748867, BD Bioscience) (1:200), GL7-Alexa Fluor 488 (Cat. 144612, Biolegend) (1:200) and PE-Klickmer-coupled RBD. The staining buffer also included BD Horizon Brilliant Stain Buffer (Cat. 563794, BD Bioscience) (1:50). 2 μL PE-Klickmer (Cat. DX01K PE 1000, Immudex) was used per sample, and it was coupled prior to use in a 1:15 molar ratio to biotinylated 6xHis- and Avi-tagged Wuhan RBD (made in-house)[85] according to the manufacturer's protocol, yielding approximately 3.5 μL total volume per sample. The PE-Klickmer-coupled RBD was added to the cells first along with 5 μL FACS buffer and was allowed to incubate for 10 min before addition of 45 μL of the premixed surface antibody staining solution, followed by incubation for 30 min. Following washing, the cells were fixated with 100 μL 4% paraformaldehyde (Cat. 47608, Sigma-Aldrich) for 10 min, washed, and finally resuspended in 350 μL FACS buffer before analysis on a FACSymphony A5 (BD Bioscience) running FACSDiva (BD Bioscience) or an Attune NxT Flow Cytometer (Invitrogen) with the Attune NxT software (Invitrogen).

## Western Blot on mucosal flushes

IgA-specific Western Blot was run on mucosal flushes to verify the presence of oligomeric forms of IgA at mucosal surfaces. Briefly, samples within a group were merged and amounts were reflected on IgA levels from FCBA. The samples were mixed in Milli-Q water to a total volume of 20 μL, containing 1X Bolt LDS Sample Buffer. For reducing gels 1 μL of DL-Dithiothreitol solution (Cat. 43816, Sigma-Aldrich) was added prior to exposure at 90 °C for 10 min. Samples were run on Bolt 4-12% Bis-Tris Plus Gel (Cat. NW04125, Invitrogen) in ice-cold 1X Bolt MES SDS Running Buffer, using Spectra Multicolor Broad Range or High Range Protein Ladders (Cat. 26623 and 26625, Thermo Scientific) as size standard. The gel was run at 80 V for 15 min followed by 110 min at 140 V while surrounded by ice blocks. After 6 min in 20% EtOH, the gel was transferred to iBlot 2 PVDF Mini Stacks (Cat. IB24002, Invitrogen) and run according to the manufacturer's instructions on an iBlot 2 Gel Transfer Device (Invitrogen). After blocking the membrane in PBS/T containing 5% BSA, IgA was detected using goat anti-mouse IgA-HRP (Cat. A90-103P, Bethyl) (1:5,000 in PBS/T 1% BSA) and SuperSignal West Pico PLUS substrate (Cat. 34580, Thermo Scientific), and subsequently imaged using GeneSys on a Syngene G:Box Chemi-XX9.

## Data analysis, graphing and illustration

HPLC data were obtained using Chromeleon Chromatography Data System Software (Version 7.2.4.0) before being graphed using GraphPad Prism software (Version 8.4.6). FCBA data were processed using Winlist 3D (Version 9.0.1, Verity Software House), and further analyses were performed in Microsoft Excel. Flow cytometry on lymph nodes were processed using FlowJo (Version 10.8.1, BD Bioscience). ELISA data were processed using Microsoft Excel. Spike pseudotyped neutralization data were processed using IncuCyte S3 (Version 2021 C, Sartorius), or BioTek Gen5 (Version 2.09, Agilent) and Microsoft Excel. SPR curves were processed using Biacore T200 Evaluation Software (Version 3.0, Cytiva). Western Blots were imaged using GeneSys (Version 1.4.1.0, Syngene). GraphPad Prism software (Version 9.0.0 or later) was used for data analysis, statistics, graphing and figure assembly. Biorender.com was used to create illustrations.

## Reporting summary

Further information on research design is available in the Nature Portfolio Reporting Summary linked to this article.

## Data availability

All data are included in the Supplementary Information or available from the authors, as are unique reagents used in this Article. The raw numbers for charts and graphs are available in the Source Data file whenever possible. Source data are provided with this paper.

## Code availability

No custom code nor mathematical algorithms were used to generate the data in this manuscript.

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

## Acknowledgements

We are grateful to Bjarne Bogen (Oslo University Hospital, Norway) who kindly shared the pLNOH2-vector with the cDNA for HA (PR8) with a C-terminal His-tag and mouse-adapted PR8 virus, and Juni Andréll (Stockholm University, Sweden) for the pCAGGS-plasmid with cDNA encoding 6xHis-tagged receptor-binding domain (RBD), to Dr. Wayne I. Lencer (Boston Children's Hospital, Harvard Medical School and Harvard Digestive Diseases Center, USA) for providing us with the HMEC-1-hFcRn cell line, and to Dr. Alexander Haas and Jens Fischer (pRED Roche Innovation Center Munich, Germany) for the MDCKII-hFcRn cell line. We are thankful to Dr. John Teijaro (The Scripps Research Institute, La Jolla, CA, USA) for kindly providing live SARS-CoV-2 (Wuhan) virus, to Siegfried Weiss (Medizinische Hochschule, Hannover, Germany) for providing the HA antibody H36-4-52, and to Davide Angeletti and Jonathan W. Yewdell (National Institute of Allergy and Infectious Diseases, National Institutes of Health, Bethesda, MD, USA) for kindly sharing the antigenic site-specific HA mAbs. We are grateful to The Unit for Research Support (Department of Pathology, Oslo University Hospital, Oslo) for preparation of HE-stained lung tissue slides. This work was partially supported by the Research Council of Norway through its Centers of Excellence scheme, project number 332727, Research Council of Norway grants 267606 (J.T.A., J.N., M.B.), 274993 (F.R.J., J.T.A., J.N., S.B.), 285136 (J.T.A., S.F., S.A.S.), 287872 (G.G., E.T.), 287927 (F.R.J., J.T.A.), 314909 (J.T.A., S.A.S.), South-Eastern Norway Regional Health Authority grants 10357 (J.T.A., F.L-J., A.K.), 2019047 (J.T.A., A.K.), 2019084 (J.T.A., A.K.A.), Independent Research Fund Denmark, Technology and Production grant 9041-00151B (D.P., K.A.H.), MRC (UK; U105181010) a Wellcome Trust Investigator Award (200594/Z/16/Z) and a Wellcome Trust Collaborator Award (214344/A/18/Z). (L.C.J., M.V.), The Coalition for Epidemic Preparedness and Innovation (CEPI) grant to monitor vaccine responses in patients on immunosuppressive therapy (F.L-J.,E.B.V., L.T.) and internal funding from Division of Head, Neck, and Reconstructive Surgery, Oslo University Hospital (T.T.G.).

## Author contributions

S.B. and E.T. contributed equally to this work. M.B., A.K.A. and A.K. performed, with assistance from S.B., H.E.L. and J.N. vector design, protein production and verification. M.B. performed in vivo vaccination and challenge studies, and serology in ELISA. A.K.A. performed HERA, in vivo vaccination and challenge studies, and serology in ELISA and FCBA, and Western Blot. S.B. performed transwell assay, in vivo vaccination studies, and assisted with serology in ELISA. A.K. performed inhibition studies of sera samples and serology in ELISA, receptor binding ELISAs and SPR. E.B.V. and L.T. performed protein conjugations and verification, and serology and inhibition of in vivo samples in FCBA. E.T. performed, with assistance from L.P., in vivo vaccination and challenge studies, RNA extraction, quantification of viral load, and pseudovirus neutralization assay on in vivo samples. A.K. and E.T. performed flow cytometry on lymph nodes. F.R.J., K.-R.J., M.L.H., S.A.S., J.N., M.D., E.T. and S.B. assisted A.K.A. with in vivo data collection at the endpoint. M.V. performed a pseudovirus neutralization assay on in vivo samples. D.P. conjugated CpG to the albumin fusion, purified, and performed quality analysis after conjugation. M.N.A. and S.A.S. assisted with protein production. A.K. and T.T.G. established the ELISA inhibition assay. S.F. assisted in vector design. B.E.L. and M.V.W. contributed to the Tg32-hFc mice strain. A.K.A. developed HE-stained lung photos, while F.L.J. analyzed and scored the HE-stained lungs. M.C.M., J.T.V., D.N., K.A.H., I.S., L.C.J., G.G., F.L-J and J.T.A. supervised the study design and preparation of the manuscript. A.K.A., A.K., E.B.V., E.T., S.B., M.B. and J.T.A. analyzed and visualized the results. A.K.A., M.B. and J.T.A. contributed to the study design and wrote the manuscript. J.T.A. contributed to forming, designing, and supervising the study and administration of the project. All authors critically revised and approved the final version of the manuscript.

## Competing interests

I.S., J.T.A. and M.B. are patent co-inventors for "Albumin Variants and Uses Thereof" (EP3063171B1, US10208102, and US10781245). I.S., J.T.A. and T.T.G. have ownership interests in Authera AS. D.P. and K.A.H. are co-inventors of patent WO2024089258A1. The remaining authors declare no competing interests.

## Additional information

[1]Department of Immunology, Oslo University Hospital Rikshospitalet, 0372 Oslo, Norway. [2]Institute of Clinical Medicine and Department of Pharmacology, University of Oslo and Oslo University Hospital Rikshospitalet, 0372 Oslo, Norway. [3]Precision Immunotherapy Alliance (PRIMA), University of Oslo, 0372 Oslo, Norway. [4]Institute of Clinical Medicine, University of Oslo, 0372 Oslo, Norway. [5]Center of Eye Research, Department of Ophthalmology, Oslo University Hospital Ullevål and University of Oslo, 0450 Oslo, Norway. [6]Protein and Nucleic Acid Chemistry Division, Medical Research Council, Laboratory of Molecular Biology, Cambridge CB2 0QH, United Kingdom. [7]Interdisciplinary Nanoscience Center (iNANO), Department of Molecular Biology and Genetics, Aarhus University, DK-8000 Aarhus C, Denmark. [8]Department of Immunology and Microbiology, The Scripps Research Institute, La Jolla, CA 92037, USA. [9]The Jackson Laboratory, Bar Harbor, ME 04609, USA. [10]Department of Pathology, Oslo University Hospital Rikshospitalet, 0372 Oslo, Norway. [11]Department of Biosciences, University of Oslo, 0371 Oslo, Norway. [12]These authors contributed equally: Anette Kolderup, Eline Benno Vaage, Malin Bern.
✉e-mail: j.t.andersen@medisin.uio.no

