## [Transparent Peer Review file · Nature Communications]

An intranasal subunit vaccine induces protective systemic and mucosal antibody immunity against respiratory viruses in mouse models

Corresponding Author: Professor Jan Terje Andersen

Version 0:

Reviewer comments:

Reviewer #1

(Remarks to the Author)

In evaluating the immunogenicity of intranasally delivered vaccine formulations based on infectious disease Ag fused to albumin, Anthi et al. demonstrate the strong induction of systemic and mucosal antigen-specific antibody responses. These responses are protective in viral challenge models. However, in fusing the Ag to albumin to increase its targeting/binding to neonatal Fc Receptor, they are also changing another important parameter that can have a large influence on immunogenicity, Ag size. The SARS-CoV-2 spike RBD domain is quite small and as such is weakly immunogenic when delivered alone, as such many other groups have fused it to a carrier to increase its size and immunogenicity. Overall, this is a generally well written manuscript with a lot of data, but for the authors to support their main conclusion that their vaccine Ag is immunogenic due to interaction of albumin with FcRn, a number of important controls need to be included.

1) Figure 1: Instead of using RBD as a control vs. RBD-MSA, a more appropriate control would be RBD-AQ and/or RBD-fused to another carrier (previously used in literature) that increases its size but not its ability to interact with FcRn. The RBD-MSA is clearly superior to RBD in inducing immune responses, but without the mentioned control, we can't be sure whether this is due to size or FcRn targeting.

2) Fig. 1k-l and Ext Data Fig 4: Here the authors are comparing apples to apples, by immunizing animals with equally sized HA Ags that either do (MSA-HA) or do not (AQ-HA) interact with FcRn. While they do show an increase in survival to viral challenge with MSA-HA, the immunogenicity data is quite weak and is not consistent with what they with the Spike RBD-based Ag. While the antigen-specific IgG and IgA titers in the serum are trending higher with MSA-HA vs. AQ-HA, they are quite inconsistent with only half the animals showing detectable titers. As such, there is no statistical difference. As for serology in the lung, there is no marked increase in antigen-specific IgA with MSA-HA over the negative control AQ-HA. I think it is critical to either increase the number of animals and/or try with a higher antigen dose to get more consistent immune responses and be able to conclude whether MSA > AQ. Finally, for panel m, the AQ-HA data is not included, and so again the comparison is between a smaller unfused HA Ag vs HA-MSA.

3) Again, for figure 2e-l, when looking in transgenic mice expressing human FcRn, they compare the RBD-QMP which should bind to the human FcR, to RBD alone, which is much smaller. If they had included the RBD-MSA they generated for Fig 1., it might have been a good control to determine how much Ag size and/or targeting are contributing to the differences in immunogenicity they are seeing.

4) Ag doses are routinely reported in micrograms. The authors refer to the molarity of antigens administered which makes it more difficult to relate to other vaccines, so I would suggest they use micrograms throughout the manuscript.

Reviewer #2

(Remarks to the Author)

In the recent study from Anthi et al, the authors report on a non-invasive vaccine platform for mucosal immunizations based on albumin-fusion proteins. This is a very simple immunization schedule, which here delivers very promising and interesting results. The idea behind the story is that antigens fused to albumin are taken up via the interaction of albumin and the FcRn, which allows an efficient uptake at the mucosal barrier.

For influenza and SARS-CoV-2, these strategy results in efficient protection against both viral infections in mice.

Furthermore, mucosal IgA responses were induced which is not seen after intramuscular immunizations with the same

fusion proteins or mRNA vaccines. The need for new formulations and strategies concerning mucosal vaccinations against respiratory viruses is definitely high as explained by the authors and mirrored by expert's opinions in the field. The presented data sets are very convincing and support the author's conclusions. They used state-of-the-art mouse models and addressed the important issue of species specificities for human or mouse FcγR or HSA/MSA. To fully demonstrate the dependence of the local immunity on FcγR binding to the HSA-proteins, one additional HSA variant would be interesting, which lost the ability to bind FcγR. However, from the transgenic mouse models there is evidence that the proposed mechanism is really important for the described immunogenicity and efficacy data. This study will be interesting for a broad readership in the field of vaccine development.

There are only minor points to consider:

- there are some duplications in the results presented in the main figures and the extended data fig., for example: Fig 1c is also part of the extended data fig.2 a and b. the same is true for Fig.1f. This should be avoided
- There are two assays describing the potential neutralizing capacity of the vaccine-induced antibodies, ACE-2 binding competition and pseudovirus assay. Both assays were performed with fixed dilutions, which does not allow the comparison of saturated samples (e.g. in Fig 2c, serum). Therefore, it would be more informative to titrate the sera/BAL and determine IC50 values for both assays. Furthermore, it is a bit confusing to describe the ACE-2 results as (% inhibition) and the pseudovirus as % infection. I would recommend to use inhibition for both assays as variable.
- the RBD-specific B-cell staining shown in ext. data 6 is not really convincing. Usually, to eliminate background staining, two fluorescently labeled antigens should be used to identify real antigen-specific B-cells. The b-cell data do not add much to the main conclusions.

Version 1:

Reviewer comments:

Reviewer #1

(Remarks to the Author)

I appreciate the authors' efforts to address my concerns by conducting new experiments with additional antigen controls. The additional data strengthens their conclusions and as such I have no further comments/concerns regarding the manuscript.

Reviewer #2

(Remarks to the Author)

In my opinion, the additional control experiments with the new carrier molecule strengthen the conclusion of this extensive study. Furthermore, my previous concerns were addressed and I see the author's points and limitations with the additional measurements, I suggested previously.

It is a well designed study and reports very interesting findings.

Point-by-point responses to reviewers' comments

Reviewer #1:

Comments for the Author:

In evaluating the immunogenicity of intranasally delivered vaccine formulations based on infectious disease Ag fused to albumin, Anthi et al. demonstrate the strong induction of systemic and mucosal antigen-specific antibody responses. These responses are protective in viral challenge models. However, in fusing the Ag to albumin to increase its targeting/binding to neonatal Fc Receptor, they are also changing another important parameter that can have a large influence on immunogenicity, Ag size. The SARS-CoV-2 spike RBD domain is quite small and as such is weakly immunogenic when delivered alone, as such many other groups have fused it to a carrier to increase its size and immunogenicity. Overall, this is a generally well written manuscript with a lot of data, but for the authors to support their main conclusion that their vaccine Ag is immunogenic due to interaction of albumin with FcRn, a number of important controls need to be included.

Comment 1: Figure 1: Instead of using RBD as a control vs. RBD-MSA, a more appropriate control would be RBD-AQ and/or RBD-fused to another carrier (previously used in literature) that increases its size but not its ability to interact with FcRn. The RBD-MSA is clearly superior to RBD in inducing immune responses, but without the mentioned control, we can't be sure whether this is due to size or FcRn targeting.

Comment 3: Again, for figure 2e-i, when looking in transgenic mice expressing human FcRn, they compare the RBD-QMP which should bind to the human FcR, to RBD alone, which is much smaller. If they had included the RBD-MSA they generated for Fig 1., it might have been a good control to determine how much Ag size and/or targeting are contributing to the differences in immunogenicity they are seeing.

Answer: We appreciate this constructive feedback and agree that the manuscript would greatly benefit from a fusion carrier that has a similar molecular weight to albumin but does not engage FcRn, so as to exclude the possibility that the strong antigen-specific antibody responses measured are mainly dependent on size. We have decided to answer question 1 and 3 simultaneously as they both relate to the same concern, performed in different mouse models.

Upon submission of our manuscript, we were already in the process of developing such a control fusion carrier similar in size to albumin. We chose to use transferrin as the alternative carrier protein. The rationale for this is that it is a soluble protein with a molecular weight (74.9 Da) not far from that of albumin (66.5 kDa). Further, it is also a ligand for a receptor expressed by mucosal epithelial cells, the transferrin receptor, which is known to recycle transferrin.

Specifically, the transferrin fusion construct was made in a similar fashion to that of the albumin antigen fusions, with the RBD antigen fused N-terminally to full-length mouse transferrin cloned into the pFUSE2ss vector backbone. To allow purification and isolation of pure monomeric fractions, a C-terminal 6xHis-tag was included. RBD-fused transferrin was well expressed in the same mammalian cell system used for the albumin-based vaccine designs. Following affinity purification on a Ni-coupled column, the eluted fractions were up-concentrated and size-exclusion chromatography was performed to isolate the monomeric fractions. RBD-fused transferrin was shown to be well expressed, and SDS-PAGE analysis revealed a pure monomeric fraction with expected molecular weight (Supplementary Fig. 7b).

Functional integrity was confirmed by ELISA, showing that the RBD-fused transferrin bound human ACE2 as well as three commercial monoclonal anti-SARS-CoV-2 IgG antibodies specific for epitopes of the spike protein (Supplementary Fig. 7c and d). In addition, binding of RBD-fused mouse transferrin to the mouse form of the transferrin receptor 1 was confirmed using SPR (Supplementary Fig. 7e). As such, we have successfully generated a functional antigen-fusion control that could be used for benchmarking against the albumin-based approach in vivo vaccine experiments.

As this manuscript contains a range of in vivo studies in three different mouse models, and several follow-up experiments are possible, we had to carefully consider how to use the colony of mice and space at our animal facility in light of the 3R's as to minimize the number of mouse studies while still addressing the key question and obtain meaningful and conclusive data. Thus, we chose to keep Fig. 1 as an introduction to the vaccine concept, while introducing the question about size-dependent immunogenicity in Fig. 2j-k. We also strategically decided to focus on studies in the state-of-the-art mouse model expressing both human FcRn and human IgG Fc (Tg32-hFc), since this model allowed us to benchmark against our lead vaccine design, namely antigen-fused human albumin with the QMP amino acid substitutions for enhanced human FcRn engagement. In contrast, BALB/c has a fully mouse background and would not be an appropriate mouse model to use as we have demonstrated that human albumin binds mouse FcRn poorly (Andersen et al., *JBC*, 2010; Bern et al., *Science TM*, 2020), and the QMP technology is only partly compensating for this distinct cross-species difference.

Following the same intranasal vaccine regimen as before (Fig. 1a and 2a), we vaccinated mice with equimolar amounts of the RBD-fused albumin and transferrin subunit proteins to directly compare the effect of the two carriers on induction of RBD-specific antibody responses. Blood, BALF and mucosal flushes were collected from the vaccinated Tg32-hFc mice at endpoint. The results showed that RBD-fused transferrin yielded no or very low antigen-specific IgG and IgA responses in any of the samples, in comparison with robust titers raised following vaccination with RBD-fused albumin-QMP (Fig. 2j). In line with this, only serum and BALF samples from RBD-QMP vaccinated mice were able to inhibit binding between ACE2 and RBD (Fig. 2k). The inclusion of RBD-Tf has been implemented in the manuscript, page 2 line 43, page 10 lines 202-215, page 13 lines 290-291, in Materials and methods page 18 lines 385-386, page 19 lines 400-401, page 23 line 519, page 28 line

628-631, page 28-29 lines 644-647 and page 29-30 lines 666-676, in figure legends page 46 lines 1108-1111, and in supplementary figure legends page 12 lines 136-143.

Comment 2: Fig. 1k-l and Ext Data Fig 4: Here the authors are comparing apples to apples, by immunizing animals with equally sized HA Ags that either do (MSA-HA) or do not (AQ-HA) interact with FcRn. While they do show an increase in survival to viral challenge with MSA-HA, the immunogenicity data is quite weak and is not consistent with what they with the Spike RBD-based Ag. While the antigen-specific IgG and IgA titers in the serum are trending higher with MSA-HA vs. AQ-HA, they are quite inconsistent with only half the animals showing detectable titers. As such, there is no statistical difference. As for serology in the lung, there is no marked increase in antigen-specific IgA with MSA-HA over the negative control AQ-HA. I think it is critical to either increase the number of animals and/or try with a higher antigen dose to get more consistent immune responses and be able to conclude whether MSA >AQ. Finally, for panel m, the AQ-HA data is not included, and so again the comparison is between a smaller unfused HA Ag vs HA-MSA.

Answer: We appreciate that the reviewer brings these issues to our attention and agree that the dataset on HA could be strengthened. To stay consistent with both the negative fusion control and the lead vaccine design used for RBD, we had also already started to construct and produce HA fused to mouse transferrin and HA fused to human albumin QMP.

As our first generation of HA-fused albumin variants gave very limited yields, we have since submission investigated strategies to improve the production yield of high-quality fractions. This was successfully achieved by truncating the C-terminal end of the HA by 22 amino acids (now amino acid 18-519) and fusing HA N-terminally to QMP-albumin (Supplementary Fig. 8a). The modified design principle was therefore extended to fusion of HA to mouse transferrin using the same vector system. Both the HA-fused vaccines were produced in the same mammalian production system as the RBD-fusions and the secreted proteins were purified as described for the respective RBD fusions, yielding high amounts of both fusion proteins. The functional integrity of the HA fusions was confirmed in ELISA showing binding to monoclonal IgG antibodies specific for HA epitopes (Supplementary Fig. 8c), while binding of mouse transferrin to the mouse transferring receptor was confirmed by SPR (Supplementary Fig. 8d) and human FcRn binding for the QMP albumin version (Supplementary Fig. 8d).

To directly compare the antigen-specific antibody responses of these functional HA-fused subunit vaccines, we again used the state-of-the-art Tg32-hFc mice for intranasal vaccination with equimolar amounts of the proteins. We followed the same vaccine regimen as before, where blood, BALF and mucosal flushes were harvested from half of the vaccinated mice 5 weeks post vaccination while we used the other half for viral challenge to measure protection.

Using ELISA, we show that HA-QMP elicited strong systemic IgG responses (Supplementary Fig. 8e), and both IgG and IgA responses in the BALF (Supplementary Fig. 8e). In stark contrast, no or very low antibody response were detected for HA-fused transferrin in serum and BALF samples (Supplementary Fig. 8e).

Importantly, when we challenged the mice with a lethal dose of Influenza A H1N1 five weeks after prime, the results demonstrated 86% survival of the HA-QMP vaccinated mice compared with only 43% for the mice given HA-fused transferrin (Supplementary Fig. 8f).

Based on these data, we have re-structured the manuscript by moving the data originally present in Fig. 1k-l and Supplementary Fig. 4 to supplementary as supporting data for the reported concept (Supplementary Fig. 10).

The new data on HA are described in the manuscript on page 10-11, lines 216-226, Materials and methods page 17-18 lines 371-377, page 19 lines 400-401, page 23 line 520 and page 29-30 lines 666-676 and supplementary figure legends pages 13-14 lines 146-161.

Comment 4: Ag doses are routinely reported in micrograms. The authors refer to the molarity of antigens administered which makes it more difficult to relate to other vaccines, so I would suggest they use micrograms throughout the manuscript.

Answer: We thank the reviewer for this suggestion. We previously chose to report doses in molarity to easier compare antigen-fused albumin with the smaller antigen alone. In light of the new data using transferrin as a non-FcRn engaging carrier control with roughly the same molecular weight as that of albumin, we have changed from molarity to micrograms in accordance with the reviewer's recommendation, in Materials and methods page 23-24, line 518-523. We have also included the vaccine doses in micrograms in the figure legends, page 45 line 1062, page 46 lines 1097-1098, 1102, 1108-1110, page 47 lines 1122-1123 and 1128, and supplementary figure legends on page 6 lines 77 and 80, page 7 lines 88 and 90, page 11 line 126, page 13 line 154, page 16 line 178, page 17 lines 191-192 and 196, page 20 line 211, page 21 lines 222-223, page 23 line 241, and page 25 line 255.

Notably, we would like to point out that the dose of our protein-based subunit vaccines can't be directly compared to the dose of the mRNA BNT162b2 vaccine used in our study, as the latter allows cells to express the encoded antigen upon vaccination.

Reviewer #2:**Comments for the Author:**

In the recent study from Anthi et al, the authors report on a non-invasive vaccine platform for mucosal immunizations based on albumin-fusion proteins. This is a very simple immunization schedule, which here delivers very promising and interesting results. The idea behind the story is that antigens fused to albumin are taken up via the interaction of albumin and the Fc γ R, which allows an efficient uptake at the mucosal barrier.

For influenza and SARS-CoV-2, these strategy results in efficient protection against both viral infections in mice. Furthermore, mucosal IgA responses were induced which is not seen after intramuscular immunizations with the same fusion proteins or mRNA vaccines. The need for new formulations and strategies concerning mucosal vaccinations against respiratory viruses is definitely high as explained by the authors and mirrored by expert's opinions in the field. The presented data sets are very convincing and support the author's conclusions. They used state-of-the art mouse models and addressed the important issue of species specificities for human or mouse Fc γ R or HSA/MSA. To fully demonstrate the dependence of the local immunity on Fc γ R binding to the HSA-proteins, one additional HSA variant would be interesting, which lost the ability to bind Fc γ R. However, from the transgenic mouse models there is evidence that the proposed mechanism is really important for the described immunogenicity and efficacy data. This study will be interesting for a broad readership in the field of vaccine development.

There are only minor points to consider:

Comment 1: There are some duplications in the results presented in the main figures and the extended data fig., for example: Fig 1c is also part of the extended data fig.2 a and b. the same is true for Fig.1f. This should be avoided

Answer: We appreciate the observant eye of the reviewer spotting the results presented in duplicates. We have now removed the duplication of serological analyses in the supplementary figures, as following:

Total IgG and IgA at endpoint in serum and BALF samples in Supplementary Fig. 2a-b, Supplementary Fig. 3a-b, Supplementary Fig. 13a, as well as chIgG1 and IgA at endpoint in serum and BALF samples in Supplementary Fig. 11a-b.

Comment 2: There are two assays describing the potential neutralizing capacity of the vaccine-induced antibodies, ACE-2 binding competition and pseudovirus assay. Both assays were performed with fixed dilutions, which does not allow the comparison of saturated samples (e.g. in Fig 2c, serum). Therefore, it would be more informative to titrate the sera/BAL and determine IC₅₀ values

for both assays. Furthermore, it is a bit confusing to describe the ACE-2 results as (% inhibition) and the pseudovirus as % infection. I would recommend using inhibition for both assays as variable.

Answer: Unfortunately, due to limited sample volume harvested from the mice, it was not possible to obtain a sigmoid curve that reached saturation for all samples. This has hindered us from confidently calculating IC₅₀ values, which is the reason we decided to report the results as % inhibition, as this allows us to compare all the analysed samples. Notably, it has been shown by Tran et al (npj vaccines, 2022) that the inhibitory effects correlate with anti-antigen titers when using the FCBA platform for analysis, further making us confident in reporting the results as % inhibition.

We appreciate the reviewer's suggestion to change the y-axis so that the pseudovirus assay and ACE2-RBD-binding data read more intuitively alongside each other. We have therefore changed the axis in Fig. 1d and g and Fig. 2c, f and k from % inhibition to % binding, and described the change in the manuscript, page 32, lines 726-728.

Comment 3: The RBD-specific B-cell staining shown in Supplementary 6 is not really convincing. Usually, to eliminate background staining, two fluorescently labeled antigens should be used to identify real antigen-specific B-cells. The b-cell data do not add much to the main conclusions

Answer: We appreciate the reviewer's input on our B cell data. However, while some researchers choose to use double positive staining of antigen-specific cells to rule out background, this is indeed just one of several rational and reliable methods for excluding background binding. Here, vaccinated mice were directly compared to PBS-treated control mice. As can be seen from Fig. 2d, there is indeed some background staining observed in the PBS-treated mice, validating the observed increase in RBD and RBD-MSA vaccinated mice. In sum, looking at the validity of experimental data, the gating strategy alone in Supplementary Fig. 4 should be evaluated alongside the control groups included in the experiment itself.